 

# Peroxiredoxin promotes longevity and H₂O₂-resistance in yeast through redox-modulation of protein kinase A

**Friederike Roger[1], Cecilia Picazo[2], Wolfgang Reiter[3], Marouane Libiad[4], Chikako Asami[1], Sarah Hanzén[1], Chunxia Gao[1], Gilles Lagniel[5], Niek Welkenhuysen[6], Jean Labarre[5], Thomas Nyström[7], Morten Grøtli[1], Markus Hartl[3], Michel B Toledano[4], Mikael Molin[1,2]***

[1]Department of Chemistry and Molecular Biology, University of Gothenburg, Gothenburg, Sweden; [2]Department of Biology and Biological Engineering, Chalmers University of Technology, Gothenburg, Sweden; [3]Mass Spectrometry Facility, Department of Biochemistry, Max F. Perutz Laboratories, University of Vienna, Vienna BioCenter, Vienna, Austria; [4]Oxidative Stress and Cancer Laboratory, Université Paris-Saclay, CEA, CNRS, Institute for Integrative Biology of the Cell (I2BC), Gif sur Yvette, France; [5]Oxidative Stress and Cancer Laboratory, Integrative Biology and Molecular Genetics Unit (SBIGEM), CEA Saclay, France; [6]Department of Mathematical Sciences, Chalmers University of Technology and University of Gothenburg, Gothenburg, Sweden; [7]Department of Microbiology and Immunology, Institute for Biomedicine, Sahlgrenska Academy, University of Gothenburg, Gothenburg, Sweden

**\*For correspondence:**
mikael.molin@chalmers.se

**Competing interests:** The authors declare that no competing interests exist.

**Abstract** Peroxiredoxins are H₂O₂ scavenging enzymes that also carry out H₂O₂ signaling and chaperone functions. In yeast, the major cytosolic peroxiredoxin, Tsa1 is required for both promoting resistance to H₂O₂ and extending lifespan upon caloric restriction. We show here that Tsa1 effects both these functions not by scavenging H₂O₂, but by repressing the nutrient signaling Ras-cAMP-PKA pathway at the level of the protein kinase A (PKA) enzyme. Tsa1 stimulates sulfenylation of cysteines in the PKA catalytic subunit by H₂O₂ and a significant proportion of the catalytic subunits are glutathionylated on two cysteine residues. Redox modification of the conserved Cys243 inhibits the phosphorylation of a conserved Thr241 in the kinase activation loop and enzyme activity, and preventing Thr241 phosphorylation can overcome the H₂O₂ sensitivity of Tsa1-deficient cells. Results support a model of aging where nutrient signaling pathways constitute hubs integrating information from multiple aging-related conduits, including a peroxiredoxin-dependent response to H₂O₂.

## Introduction

Caloric restriction (CR) is an intervention that slows down aging and reduces the incidence of age-related disease from the unicellular baker's yeast (*Lin et al., 2000*) to rhesus monkeys (*Mattison et al., 2017*). CR-induced reduced nutrient signaling via insulin/insulin-like growth factor (IGF-1), the target-of-rapamycin and/or protein kinase A pathways is intimately linked to lifespan extension (*Fontana et al., 2010*; *Kenyon, 2010*; *Molin and Demir, 2014*; *Nyström et al., 2012*). Of other things, reduced nutrient signaling mitigates age-related oxidative damage by increasing oxidative stress resistance in organisms from yeast to humans (*Fontán-Lozano et al., 2008*; *Heilbronn et al., 2006*; *Molin et al., 2011*; *Schulz et al., 2007*; *Sohal and Forster, 2014*). Increased

oxidative stress resistance appears as a common denominator of mechanisms by which nutrient signaling pathways dictate the anti-aging effects of CR and its health benefits (*Alic and Partridge, 2011*; *Fontana et al., 2010*; *Longo et al., 2012*). Still very few specific targets of nutrient signaling that explain the beneficial effects of CR have been identified (*Fontán-Lozano et al., 2008*).

Peroxiredoxins might constitute one such target, as this major family of peroxide-negating enzymes is required for lifespan promotion by CR and CR-mimetics (*De Haes et al., 2014*; *Molin et al., 2011*; *Oláhová and Veal, 2015*). In worms, the CR-mimetic drug metformin extends lifespan in a manner dependent on the activity of Prdx-2 (*De Haes et al., 2014*), and in flies, neuronal peroxiredoxin overexpression extends lifespan in the absence of caloric restriction (*Lee et al., 2009*). In addition, CR increases both yeast $H_2O_2$ tolerance and lifespan by stimulating the activity of the major 2-Cys peroxiredoxin, Tsa1 (*Molin et al., 2011*), and the mild overexpression of Tsa1 potently extends lifespan by 40% (*Hanzén et al., 2016*). As peroxiredoxins have been described as major peroxide scavenging enzymes, they may reduce the rate of aging by scavenging $H_2O_2$, which may also explain their requirement for the maintenance of genome stability (*Molin and Demir, 2014*; *Nyström et al., 2012*) and the premature accumulation of age-related tumors in PrxI-deficient mice (*Neumann et al., 2003*). However, mild Tsa1 overexpression, although increasing lifespan, did not alter the rate at which mutations accumulate during aging (*Hanzén et al., 2016*). Furthermore, CR reduced the increased mutation rate in Tsa1-deficient cells by 50% (*Hanzén et al., 2016*) without extending their life-span (*Molin et al., 2011*). We instead proposed that Tsa1 counteracts age-related protein damage by guiding Hsp70/104 molecular chaperones to proteins aggregating upon increased age and $H_2O_2$ (*Hanzén et al., 2016*).

Prx are obligate dimers carrying two catalytic residues, the peroxidatic Cys ($C_P$, Cys48 in Tsa1) and the resolving Cys ($C_R$, Cys171 in Tsa1). $C_P$ reduces $H_2O_2$ and forms a sulfenic acid (-SOH), which condenses with the $C_R$ of the second Prx molecule into an inter-subunit disulfide, then reduced by thioredoxin. Once formed, the $C_P$-SOH can also react with another $H_2O_2$ molecule, which leads to formation of a sulfinic acid (-SO$_2$H), instead of condensing into a disulfide. Sulfinylation inactivates the catalytic cycle, switching the enzyme function into a molecular chaperone by multimerisation (*Hanzén et al., 2016*; *Jang et al., 2004*; *Noichri et al., 2015*). Prxs can also signal $H_2O_2$ by transfer of the oxidant signal to target proteins (*Leichert and Dick, 2015*; *Stöcker et al., 2018a*).

We recently showed that, in response to $H_2O_2$, Tsa1 and thioredoxin are required for the activation of the transcription factor Msn2, as it inhibits PKA-mediated Msn2 repression (*Bodvard et al., 2017*). Here, we explored whether the modulation of PKA by Tsa1 had any relevance in its role in slowing down aging and in $H_2O_2$ resistance. We show that both the premature aging and $H_2O_2$ sensitivity of cells lacking Tsa1 is due to aberrant protein kinase A (PKA) activation, and not to defective $H_2O_2$ scavenging per se. Similarly, a single extra copy of the *TSA1* gene extended life-span by mildly reducing PKA activity, without affecting $H_2O_2$ scavenging. Tsa1 interacts with PKA at the level of its catalytic subunits. We identified a conserved Cys residue in the PKA catalytic subunit Tpk1 that is specifically required for Tsa1-mediated $H_2O_2$ resistance. Tsa1-dependent oxidation of the catalytic subunit reduced enzyme activity and increased $H_2O_2$ resistance in part through dephosphorylating a conserved threonine (Thr241) in the kinase activation loop. These results indicate that peroxiredoxins slow down the rate of aging through a unique role in kinase signaling, in addition to promoting proteostasis. They also suggest a novel mode of regulation of the conserved nutrient-sensing cascade PKA that bypasses conventional signaling via the second messenger cAMP, and impinges on both $H_2O_2$ resistance and aging.

## Results

### The effects of Tsa1 on longevity are mediated by the Ras-cAMP-PKA pathway

A single extra-copy of the *TSA1* gene, which encodes the major yeast cytosolic Prx, Tsa1, prolongs lifespan in the absence of caloric restriction (*Hanzén et al., 2016*). To clarify the mechanism by which Tsa1 promotes this effect, we enquired whether PKA is involved, as this kinase antagonizes both longevity (*Lin et al., 2000*) and resistance to $H_2O_2$ (*Molin et al., 2011*) and Tsa1 is required for decreasing PKA-dependent phosphorylation of the 'general stress' transcription factor Msn2 in response to $H_2O_2$ (*Bodvard et al., 2017*). The high affinity cAMP-phosphodiesterase Pde2 degrades cAMP, and

deletion of *PDE2* promotes PKA activation by increasing cAMP levels, downstream of Ras2 (*Figure 1A*; *Broach, 2012*; *Deprez et al., 2018*; *Santangelo, 2006*). Deletion of *PDE2* decreased the lifespan of the wild type strain by 45% (*Figure 1B*), as previously shown (*Lin et al., 2000*), and also prevented the increased lifespan conferred by mild overexpression of *TSA1* (compare *pde2Δ* and *pde2Δ o/e TSA1*), which indicates that PKA activity is dominant over Tsa1, and suggests that Tsa1 might slow down aging by decreasing PKA activity. Indeed, mild *TSA1* overexpression increased both the accumulation of the reserve carbohydrate glycogen (*Figure 1C*), a diagnostic feature of low PKA activity, and the expression of the PKA-repressed Msn2/4 target Hsp12 (*Figure 1D*).

We turned to cells lacking *TSA1*, which suffer a severely shortened lifespan (*Molin et al., 2011*), asking whether this phenotype is linked to PKA. We combined the deletion of *TSA1* and *RAS2*, the latter largely abrogating the stimulation of PKA by glucose (*Figure 1A*; *Santangelo, 2006*). Strikingly, Ras2 deficiency completely rescued the reduced lifespan (*Figure 1E*) of cells lacking Tsa1, and upon deletion of *PDE2* in these cells (*ras2Δtsa1Δpde2Δ*), this rescue was no longer visible (*Figure 1F*). These data indicate that the shortened lifespan of *tsa1Δ* is due to aberrant activation of the Ras-PKA pathway, and as a corollary, that Tsa1 might inhibit this pathway. That Tsa1 deletion did not further reduce the lifespan of Pde2-deficient cells (*Figure 1F*), further support the notion that Tsa1 influences longevity by repressing the Ras-PKA pathway.

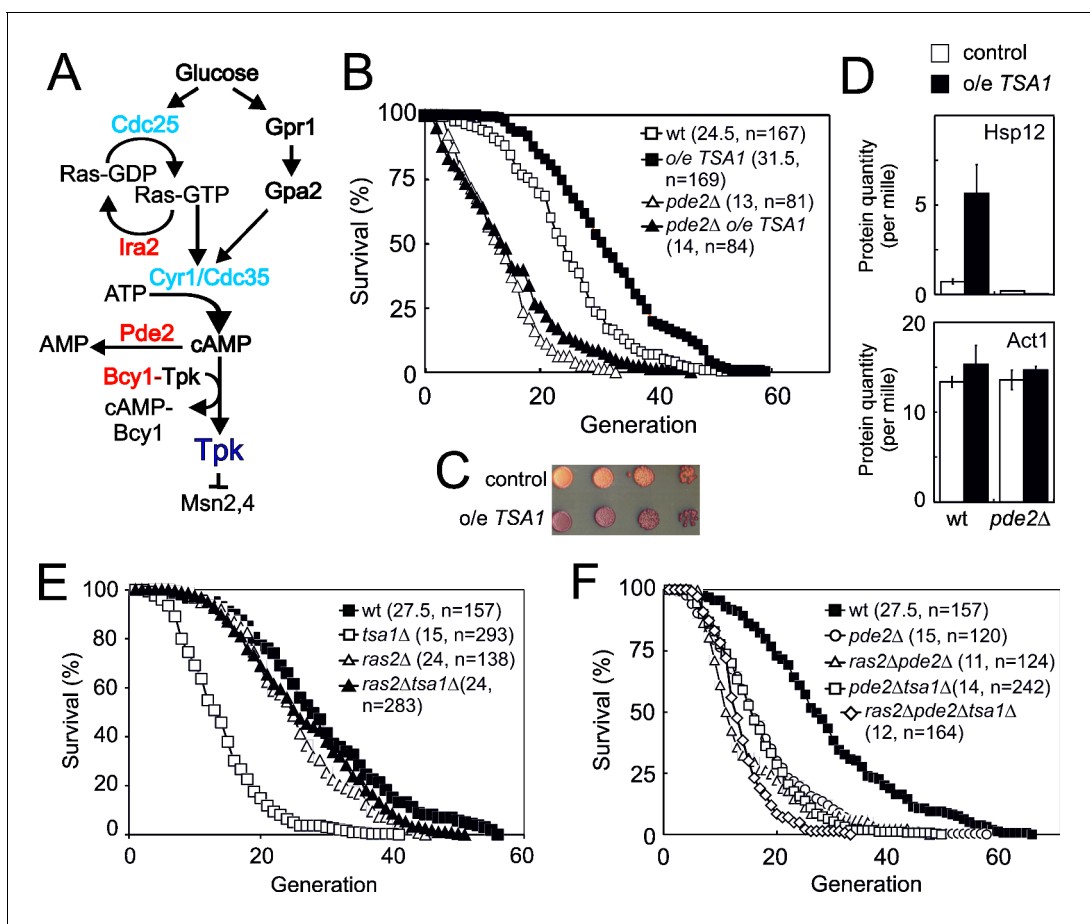

**Figure 1.** The 2-cys peroxiredoxin Tsa1 slows down aging via inhibiting protein kinase A signaling. (**A**) Overview of the Ras-cAMP-PKA signaling pathway. In blue stimulatory components and in red inhibitory. (**B**) Lifespans of cells expressing an extra copy of the *TSA1* gene or not (vector control) in combination with the deletion of *PDE2* to induce high PKA signaling (*pde2Δ*). (**C**) Accumulation of glycogen in vector control cells or cells expressing an extra copy of the *TSA1* gene as assayed by iodine vapor. (**D**) Expression of Hsp12 in the indicated mutant strains (n = 3). (**E–F**) Lifespan of cells lacking Tsa1, Ras2, Pde2 or combinations thereof.

## Tsa1 represses the Ras-cAMP-PKA pathway at the level of the PKA enzyme

Cells lacking Ras2 grew significantly slower than the wild-type (*Figure 2A*), consistent with a substantial reduction in PKA activity. However, deleting *TSA1* in these cells (*ras2Δtsa1Δ*) rescued their slow growth to a rate indistinguishable from that of *tsa1Δ* cells (*Figure 2A*), again pointing to an antagonistic effect of Tsa1 on the Ras-PKA pathway, also suggesting that Tsa1 affects the pathway downstream of Ras2. Similarly, overexpressing Ira2, a Ras-GTPase activating protein (RasGAP) that decreases PKA activation by switching RAS-GTP to its inactive GDP form, both slowed down growth to approximately half the rate of control cells (*Figure 2C*) and increased expression of Msn2/4-target genes that are under PKA repression (*Figure 2D*). Deleting *TSA1* in this strain restored both phenotypes (*Figure 2B–D*), similar to the effect of Ras-overactivation (*RAS2G19V* allele, *Figure 2—figure supplement 1A–B*) or Pde2 deficiency (*Figure 2C*). Importantly, rescue of the slow growth of Ira2-overproducing cells by Tsa1 deletion was lost when *PDE2* was also overexpressed in these cells, also indicating that the rescue is due to increased PKA activity (*Figure 2E*).

Strains lacking both *RAS* alleles (*ras1Δras2Δ*) are not viable due to inactivation of PKA. This inability to germinate can be rescued by genetic interventions that restore PKA activity downstream of Ras, i.e. the inactivation of Pde2 or of the PKA negative regulatory subunit Bcy1 (*Garrett and Broach, 1989*; *Toda et al., 1985*; *Wilson and Tatchell, 1988*), or of Yak1, which acts downstream of PKA. Loss of the PKA-repressed 'general stress' transcription factor Msn2 alone or of both Msn2 and its homologue Msn4, can also partially overcome the growth impairment of the partial loss of active Ras (*Figure 2C*) and the requirement for a PKA catalytic subunit-encoding (*TPK*)-gene for viability (*Smith, 1998*). As the above data indicate that Tsa1 represses PKA activity, we tested whether its loss could similarly rescue the inability of *ras1Δras2Δ* to germinate by sporulating heterozygous *ras1Δ/RAS1*, *ras2Δ/RAS2* and *tsa1Δ/TSA1* diploid cells. However, no cells lacking both Ras1 and Ras2 were viable irrespective of the presence or absence of Tsa1 (*Figure 2F*). Similarly, we did not obtain viable *tsa1Δ tpk1Δ tpk2Δ tpk3Δ* spores in a cross between haploid *tsa1Δ* and *tpk1Δ tpk2Δ tpk3Δ* strains unless a centromeric *TPK1* plasmid was also present (in 6 out of 6 viable spores with the genomic *tsa1Δ tpk1Δ tpk2Δ tpk3Δ* genotype (*Figure 2—figure supplement 1C*). These data suggest that the repression exerted by Tsa1 on the Ras-PKA pathway requires the presence of PKA, and thus that the latter may be the target of repression.

To further ascertain at which level Tsa1 interferes with Ras-cAMP-PKA activity, we overproduced the PKA negative regulatory subunit (mc-*BCY1*), which by inactivating PKA releases repression of Msn2, and dramatically increases the latter's response to $H_2O_2$ (*Bodvard et al., 2017*; *Figure 2G*). However, mc-*BCY1* had no effect in Tsa1-deficient cells (*Figure 2G*), suggesting that Tsa1 inhibits the Ras-cAMP-PKA pathway at the level of the PKA enzyme. We also measured the levels of the pathway signaling intermediates, Ras-GTP and cAMP, in cells overproducing Ira2 in the presence and absence of Tsa1. As expected, overexpression of *IRA2* dramatically reduced the levels of active Ras (Ras-GTP) and this reduction was largely maintained in *pde2Δ* cells (*Figure 2H*), in which PKA signaling is increased downstream of Ras. Similarly, Tsa1-deficient cells overproducing Ira2 exhibited very low Ras-GTP levels (*Figure 2H*). In addition, cAMP levels were not affected in Tsa1-deficient cells (*Figure 2I*). Altogether, these data indicate that repression of the Ras-cAMP-PKA pathway by Tsa1 is needed both during aging and normal growth, and that this repressive effect is exerted at the level of PKA. Lastly, to directly monitor the impact of Tsa1 on PKA activity, we used a PKA sensor in which the phosphorylation state of the ectopic PKA site LLRAT*-LVD in the mammalian FHA1 phospho-amino acid domain is evaluated via FRET (*Molin et al., 2020*). PKA repression upon $H_2O_2$ addition was readily visible in wild-type cells using this sensor, whereas cells lacking Tsa1 hardly repressed PKA at all (*Figure 2J*).

Altogether, these data indicate that repression of the Ras-cAMP-PKA pathway by Tsa1 is exerted at the level of PKA, and occurs during aging, in the cell response to $H_2O_2$ and during normal growth.

## Tsa1 catalytic cysteines control $H_2O_2$ resistance by repressing PKA

Prxs can function as $H_2O_2$ scavengers, as receptors of $H_2O_2$ signaling relays, or as chaperones. The first two functions require Prx-two catalytic Cys residues $C_P$ and $C_R$ and electrons from thioredoxin, whereas the third one only relies on the sulfinylation of $C_P$. To sort out which of these three Prx



**Figure 2.** The Tsa1 catalytic cysteines affect protein kinase A dependent proliferation downstream of cAMP but not downstream of the catalytic subunits. (**A**) Growth of cells lacking Ras2, Tsa1 or both (n = 3, error bars indicate SD). (**B–C**) Growth of cells overexpressing *IRA2* in the indicated mutants of the Tsa1 catalytic cycle or the PKA signaling pathway on solid (**B**) or in liquid medium (**C**), n = 3–15. (**D**) Expression of the PKA repressed *CTT1* or *HSP12* genes in the indicated mutants in the Tsa1 catalytic cycle overexpressing *IRA2* (mc-*IRA2*) or not (instead expressing the vector, control, n = 3 ± SD) sampled during mid-exponential growth. (**E**) Growth of Tsa1-proficient or deficient (*tsa1Δ*) cells overexpressing *IRA2* (mc-*IRA2*) or *PDE2* (mc-*PDE2*), both or the corresponding vector control plasmids (control) in liquid medium (n = 3 ± SD). (**F**) Spore germination in cells deficient in Ras1, Ras2, Tsa1 or combinations thereof. Spore germination was estimated in 32 tetrads where genotypes could be assigned to all spores (128 in total, 8–23 spores per genotype). (**G**) Total time of nuclear Msn2 localization in the indicated mutant strains for 60 min following the addition of 0.3 mM H₂O₂ (n = 46–82). (**H–I**) Ras-GTP (**H**) or cAMP (**I**) levels in the wild-type or the indicated mutant strains overexpressing *IRA2* (mc-*IRA2*) or not (expressing the vector control, control, n = 3). (**J**) Phosphorylation of the ectopic AKAR4 PKA site upon H₂O₂ addition (0.4 mM) in wt, *tsa1Δ* and *trx1Δtrx2Δ* cells. (n = 85, 71 and 32, respectively). Error bars indicate SD.

The online version of this article includes the following figure supplement(s) for figure 2:

**Figure supplement 1.** Tsa1 and the cytosolic thioredoxins Trx1 and Trx2 impact on PKA related growth signaling but lack of Tsa1 cannot overcome the requirement for a PKA catalytic subunit for spore viability.

biochemical functions is involved in PKA repression, we examined the effect of mutating $C_P$ and $C_R$ or of preventing enzyme sulfinylation on Tsa1-mediated repression. The lifespans of $tsa1C48S$ and $tsa1C171S$ mutants suffered a lifespan as short as cells lacking Tsa1 (*Figure 3A*). Similarly, both the slow growth and the constitutive expression of the PKA-repressed genes *CTT1* and *HSP12* resulting from Ira2 overproduction were lost in the $tsa1C48S$ and $tsa1C171S$ mutants (*Figure 2B–D*). In contrast, cells expressing a truncated form of Tsa1 lacking the C-terminal YF motif ($tsa1\Delta YF$), an enzyme form almost totally resilient to sulfinylation (*Hanzén et al., 2016*), were indistinguishable from wild-type with regards to their lifespan (*Figure 3A*), slow growth and Ira2 overexpression-dependent, constitutive Msn2-target expression (*Figure 2B–D*), thus excluding an involvement of the Tsa1 chaperone function in PKA repression.

Next, to differentiate between the scavenging and signaling functions of Tsa1, we first probed the $H_2O_2$ sensitivity phenotype of cells lacking Tsa1. The $tsa1\Delta$ was sensitive to $H_2O_2$, as monitored

**Figure 3.** Tsa1 catalytic cysteines slow down aging and increase $H_2O_2$-resistance via inhibiting protein kinase A. (**A**) Life spans of wild-type or the indicated genomic *tsa1* mutant strains. In brackets median life-spans and n. (**B**) Spot-test assay of growth in the presence and absence of 1.5 mM $H_2O_2$ in YPD plates. (**C**) Quantification of $H_2O_2$ resistance in (**B**) (n = 3). (**D**) $H_2O_2$ resistance (1.5 mM $H_2O_2$, YPD medium) in the indicated mutants (n = 3). (**E**) $H_2O_2$ resistance in cells overexpressing *IRA2* (mc-*IRA2* +) or vector control (-) 0.4 mM $H_2O_2$, SD medium (n = 3). (**F**) Culture medium $H_2O_2$ removal assay of wt (black) and *tsa1Δ* cells (blue) to which 200 µM was added. Inset shows average scavenging rates for cultures upon the addition of 400 µM (n = 3). Error bars indicate SD. (**G**) Average HyPer3 (red) or HyPer3 C199S (black) fluorescence ratio (500 nm/420 nm) in young or aged wild-type or *tsa1Δ* cells +/- 400 µM $H_2O_2$ for 10 min. Cells of about 10–12 generations of replicative age (aged) or young control cells (young) were assayed. Error bars indicate SEM (n = 231, 170, 319, 236 and 202, respectively). (**H**) Average HyPer3 (red) or HyPer3 C199S (black) fluorescence ratio (500 nm/420 nm) in young or aged wild-type (YMM130) and o/e *TSA1* cells as in (**G**) Error bars indicate SEM (n = 404, 579, 190 and 204, respectively).

The online version of this article includes the following figure supplement(s) for figure 3:

**Figure supplement 1.** Reduced Ras activity can overcome $H_2O_2$ sensitivity of cells lacking Tsa1 but not that of cell lacking the cytosolic thioredoxins Trx1 and Trx2.

by growth on plates containing $H_2O_2$, and strikingly, deletion of *RAS2* or the overproduction of Ira2 totally rescued this phenotype (*Figure 3B–C*). Deletion of *PDE2* in these cells (*ras2Δtsa1Δpde2Δ* or *pde2Δtsa1Δ* mc−*IRA2*) restored the $H_2O_2$ sensitivity of *tsa1Δ* (*Figure 3E*; *Figure 3—figure supplement 1A*), further indicating that the *tsa1Δ* $H_2O_2$ phenotype is linked to overactive PKA, and not to the loss of Tsa1 scavenging function. Similarly, mild overexpression of *TSA1* conferred an increased tolerance to $H_2O_2$, which was lost upon deletion of *PDE2* (*Figure 3D*). As another indication of Tsa1 scavenging function dispensability, the decay rate of $H_2O_2$ in the medium of *tsa1Δ* cells after adding a bolus dose was similar to the rate observed in a wild-type cell suspension (*Figure 3F*). In addition, $H_2O_2$ levels measured using the genetically encoded $H_2O_2$ sensor HyPer3 (*Bilan et al., 2013*) were modestly, but significantly increased in old wild-type (10–12 generations), relative to young cells (*Figure 3G*). Tsa1-deficient cells however, exhibited a similar or even lower increase in the $H_2O_2$ fluorescence ratio with age, relative to wild-type, and in cells expressing an extra copy of the *TSA1* gene, $H_2O_2$ increased to a similar or even higher levels in aged cells (*Figure 3H*).

We also examined the role of the thioredoxin pathway in PKA repression, which although required for both Tsa1 signaling and scavenging functions, should be more important for the latter. Deletion of *TRX1* and *TRX2* partly rescued the slow growth of *IRA2*-overexpressing cells (*Figure 2B–C*), and suppressed the increased constitutive expression of the PKA-repressed Msn2/4 target genes resulting from Ras2 deletion (*Figure 2—figure supplement 1D*), the latter even more so than did the deletion of *TSA1*. However, although $H_2O_2$ sensitive, this *trx1Δtrx2Δ* strain $H_2O_2$ phenotype could neither be rescued by deletion of *RAS2* (*Figure 3—figure supplement 1B*) nor by the overproduction of Ira2 (*Figure 3E*). In addition, PKA was still moderately repressed in *trx1Δtrx2Δ* in response to $H_2O_2$, as measured with the FRET PKA phosphorylation sensor (*Figure 2J*). Thioredoxins are thus only partially required to repress the phosphorylation of an ectopic PKA target site upon $H_2O_2$ addition, or may govern signaling through another pathway that synergizes with PKA in some PKA output functions. Nevertheless, that the Tsa1 catalytic Cys residues are critical to restrain PKA activity, but not the thioredoxins further exclude the Tsa1 scavenging function per se.

## Tpk1 is sulfenylated upon $H_2O_2$ addition and glutathionylated on the conserved Cys243

If indeed Tsa1 inhibits PKA, we asked by which mechanism this happens. We detected in myc-Tsa1 immunoprecipitates from unstressed cells a weak, but significant amount of Tpk1, the amount of which increased dramatically following $H_2O_2$ addition (0.4 mM, *Figure 4A*). Moreover, immunoprecipitating Tpk1-HB brought down a significant amount of Tsa1 (*Figure 4—figure supplement 1A*). We next asked whether PKA underwent thiol-redox modifications. Non-reducing electrophoresis did not identify any migration changes compatible with the presence of a disulfide in neither of Tpk1 nor Bcy1 (*Figure 4—figure supplement 1B–C*). Similarly, kinetic-based trapping using *tsa1-* and *trx2*-resolving cysteine mutants (*tsa1C171S* and *trx2C34S*) neither altered Bcy1 nor Tpk1 migration (*Figure 4—figure supplement 1B–D*). We thus performed a mass spectrometry (MS) analysis using affinity-purified His-biotin-tagged Tpk1 (Tpk1-HB) (*Tagwerker et al., 2006*; *Supplementary file 1A*). We first performed shot-gun MS and open search analysis to determine the most abundant Tpk1 PTMs of its two Cys residues, Cys195 and Cys243, followed by a targeted label-free quantification approach on a selected set of peptides using parallel reaction monitoring (PRM). A significant proportion of Cys195 was present as an adduct with glutathione (GSH) in unstressed cells (*Figure 4—figure supplement 1E*, *Supplementary file 1B*), and levels of all three peptides bearing this modification decreased by 6 and 11-fold upon cell exposure to 0.4 mM and 0.8 mM $H_2O_2$, respectively (*Figure 4—figure supplement 1F*, *Supplementary file 1C-D*). A significant fraction of Tpk1 Cys243 was also glutathionylated, even in unstressed cells, and in this peptide, Thr241 was phosphorylated (*Figure 4B–C*, *Figure 4—figure supplement 1G–H*, *Supplementary file 1B*). We also detected variants of this peptide bearing other cysteine modifications (i.e. methyl thiolation, sulfinylation and unknown modifications, *Figure 4—figure supplement 1G*). Importantly, Thr241 phosphorylation decreased upon exposure to $H_2O_2$ (*Figure 4C*, *Figure 4—figure supplement 1G*), as did Cys243 glutathionylation (2.5-fold), when it occurred on the phosphorylated peptide (*Figure 4C*). However, the Cys243 glutathionylated Thr241 dephosphorylated peptide increased by 1.4 fold. Confirming MS results, Tpk1 was glutathionylated in unstressed cells, when monitored by anti-glutathione immunoblot of immunoprecipitated Tpk1-HB (*Figure 4E–F*), and this signal decreased upon exposure to $H_2O_2$. Further, in *tsa1Δ* cells, the glutathionylation signal was more intense, and did not decrease,

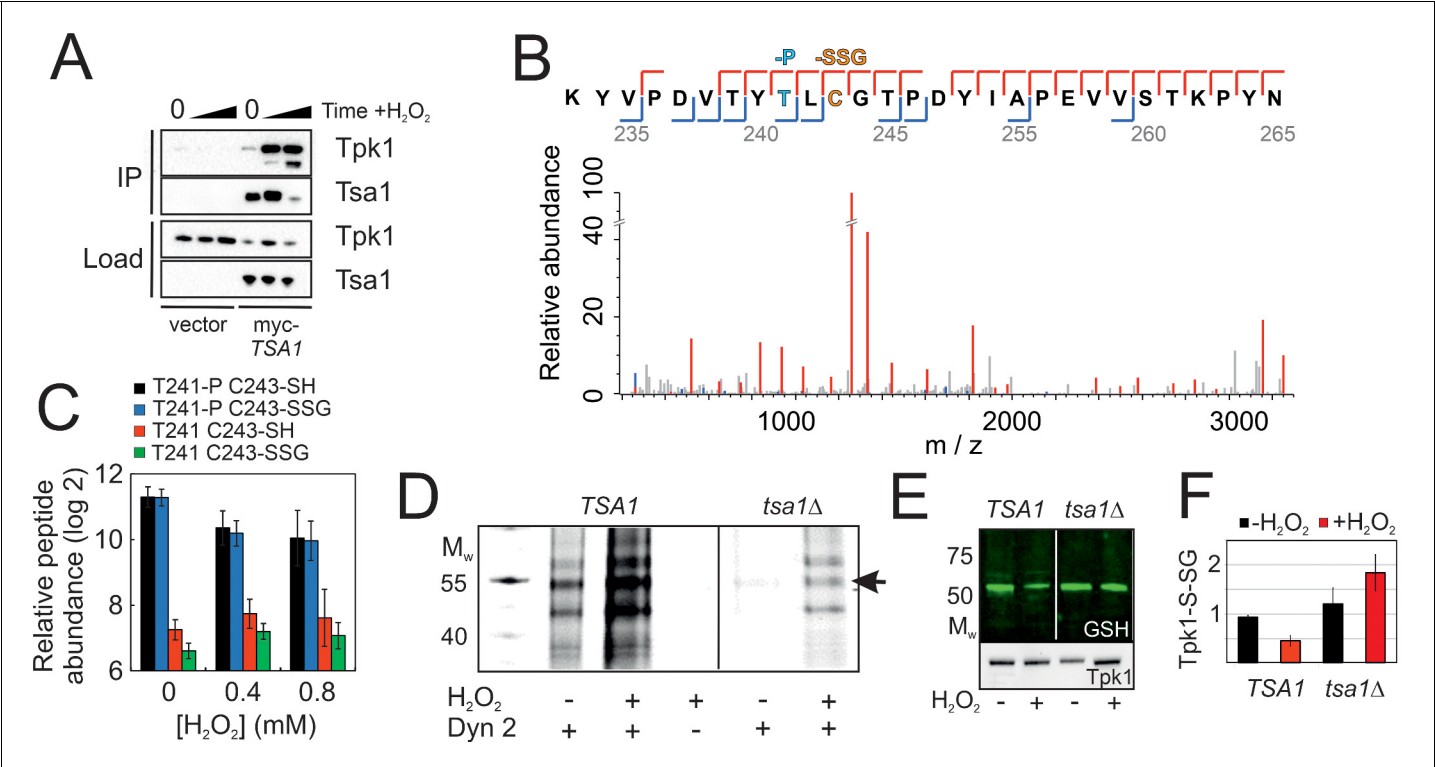

**Figure 4.** Tsa1 interacts with the PKA catalytic subunit Tpk1 and stimulates Tpk1 cysteine sulfenylation by $H_2O_2$. Tpk1 is glutathionylated at a conserved cysteine. (**A**) Tpk1 interacts with myc-Tsa1 in a coimmunoprecipitation assay and in a manner strongly stimulated by $H_2O_2$. (**B**) MS-MS spectrum showing the matching b-ion (blue) and y-ion (red) series following fragmentation of the Thr241 phosphorylated and C243 glutathionylated peptide encompassing amino acid residues Y239-K261 in Tpk1. T-P = phospho threonine, C-SSG = glutathionylated cysteine. (**C**) PRM-based quantification of the indicated Thr241 and Cys243 containing Y239-K261 peptides in Tpk1, in the absence or presence of the indicated amount of $H_2O_2$, respectively (n = 3). Error bars indicate SD. (**D**) DYn-2 assay showing Tpk1 cysteine sulfenylation in the presence and absence of *TSA1* and +/- 0.5 mM $H_2O_2$ for 5 min. Tpk1-HB was immunoprecipitated from *tpk2Δtpk3Δ* (*TSA1*) and *tpk2Δtpk3Δtsa1Δ* (*tsa1Δ*) cells and analyzed in gel for cyanine5 fluorescence. (**E–F**) Glutathionylation of Tpk1-HB in strains in (**D**) as assayed by anti-glutathione immunoblot of immunoprecipitated Tpk1-HB in the absence of or 10 min following the addition of 0.4 mM $H_2O_2$. Extracts were separated under non-reducing conditions (n = 3).
The online version of this article includes the following figure supplement(s) for figure 4:

**Figure supplement 1.** Tsa1 interacts with the PKA catalytic subunits Tpk1, controls Tpk1 cysteine sulfenylation independent on disulphide formation and a significant proportion of Tpk1 cysteines are glutathionylated under basal conditions.

but rather increased upon $H_2O_2$ exposure. We also used DYn-2, a cell-permeable cysteine sulfenic acid (-SOH) probe amenable to click chemistry (*Yang et al., 2015*) as another approach to probe Tpk1 oxidative modifications. In wild-type unstressed cells, Tpk1 displayed a weak DYn-2 signal, the intensity of which significantly increased upon $H_2O_2$ addition, whereas in *tsa1Δ* cells this signal was much less intense, both prior to and after exposure to $H_2O_2$ (0.4 mM, *Figure 4D*, *Figure 4—figure supplement 1I*).

The two Tpk1 Cys residues thus undergo complex redox changes comprising glutathionylation and sulfenylation as dominant and biologically relevant modifications. These changes occur independently, the former present in unstressed cells, decreasing upon $H_2O_2$ exposure, and the latter strongly induced by $H_2O_2$, both dependent upon Tsa1. The fact that the increased sulfenylation of Tpk1 upon $H_2O_2$ addition correlates with Thr241 dephosphorylation led us to probe the importance of all three residues in Tpk1 function by Ala substitution. These substitutions altered neither protein levels nor the ability of cells to grow (*Figure 5—figure supplement 1A–D*). Interestingly, *tpk1C243A*, but not *tpk1C195A* rendered cells hyper-sensitive to $H_2O_2$ (*Figure 5A*, *Figure 5—figure supplement 1E*), which was not improved by mild overexpression of *TSA1* (*Figure 5B*). In contrast, the *tpk1T241A* mutant significantly increased $H_2O_2$ resistance both in wild-type (*Figure 5A*) and in *tsa1Δ* cells (*Figure 5C*). A docking experiment performed on a Tpk1 3D structural homology



**Figure 5.** Tpk1 Cys243 redox-modification and Tsa1 inhibits PKA activity by dephosphorylating and destabilizing the activation loop. (**A–B**) $H_2O_2$ resistance of the wild-type vector control (**A**, pRS313 or **B**, pRS403) or the indicated *tsa1-* or *tpk*-mutant strains in SD medium, 0.6 mM $H_2O_2$. Strains in (**B**) carry pRS316-*TPK1* or pRS316-*tpk1C243A* as the only PKA catalytic subunit peroxiredoxin Tsa1 slows down (genomic *tpk1Δtpk2Δtpk3Δ* deletions, n = 3). (**C**) $H_2O_2$ resistance of *tpk1Δtpk2Δtpk3Δ* and *tpk1Δtpk2Δtpk3Δtsa1Δ* cells transformed with pRS313-*TPK1* or pRS313-*tpk1T241A* as indicated in SD medium 0.6 mM $H_2O_2$ (n = 3). (**D–E**) Structural homology model of yeast Tpk1 (**D**) based on the structure of mouse type II PKA holoenzyme (**E**) [PDB ID 3TNP, (*Zhang et al., 2012*). (**F–I**) Amino acids in the activation loop (in red) of Tpk1 in the Thr241 phosphorylated Cys243 non-modified (**F**), Thr241 non-phosphorylated Cys243 non-modified (**G**), Thr241 non-modified Cys243 glutathionylated (**H**) and Thr241 phosphorylated Cys243 glutathionylated (**I**) states in the Tpk1 structural homology model. The backbones are colored in light blue, carbon atoms in yellow, nitrogen atoms in blue, oxygen atoms in red and phosphor atoms in scarlet. The distance between Lys233 and phosphorylated Thr241 is 9.55 Å (**F**) whereas Lys233 and non-phosphorylated Thr241 reside 10.88 Å apart (**G**). (**J**) Overview of mechanisms by which glucose and $H_2O_2$ control PKA activity. In blue activators and in red inhibitors. See also *Figure 5—figure supplement 1*.

The online version of this article includes the following figure supplement(s) for figure 5:

**Figure supplement 1.** Substitution of Cys195, Thr241 and Cys243 by alanine in the yeast.

model based on the mouse enzyme structure (*Figure 5D–E*), showed that introducing a glutathione moiety at Cys243 stabilized Thr241 in the dephosphorylated state by direct hydrogen bonding (*Figure 5F–H*). When Thr241 was phosphorylated, the kinase activation loop was now stabilized through hydrogen bonds to Arg209 and Lys233 (*Figure 5F–G*), and in this setting, glutathione at Cys243 adopted a different position, now extending towards the ATP-binding pocket (*Figure 5I*). Subsitution of Cys243 to the less bulky cysteine sulfenic/sulfinic acid mimetic aspartate (*tpk1C243D*), or modification by methylthiolation (S-CH₃, *Figure 4—figure supplement 1E*) had, however, little effect on the molecular dynamics of Tpk1 (*Figure 5—figure supplement 1F–G*). In summary, Cys243 glutathionylation might inhibit PKA by interfering both with Thr241 phosphorylation and with the ATP-binding pocket dynamics, when occurring together with phosphorylated Thr241, which would not fit the observed decreased glutathionylation of Tpk1 seen upon $H_2O_2$ addition. Alternatively, the Cys243 sulfenic acid may react further as previously speculated for the redox modulation

**Figure 6.** Model of the mechanisms by which altered peroxiredoxin levels impacts on aging. In the first mechanism peroxiredoxin-dependent redox-signaling impacts in an unconventional manner on the PKA nutrient signaling kinase (this study) and in the other on proteostasis (*Hanzén et al., 2016*). (A) In wild-type cells Tsa1 catalytic cycling maintains longevity by decreasing PKA-dependent nutrient signaling leading to the stimulation of maintenance but at the expense of growth. (B) In cells lacking Tsa1, nutrient signaling is aberrantly increased leading to reduced maintenance and increased growth. (C) Enforced expression of the peroxiredoxin Tsa1 slows down aging both by repressing nutrient signaling (this study) and by stimulating protein quality control mechanisms to reduce the levels of damaged and aggregated protein (*Hanzén et al., 2016*).

of the ER kinase IRE-1 (*Hourihan et al., 2016*) and our 3D data suggest that a more bulky modification may be the driving event in PKA repression. Taken together, these data support the presence of a Tsa1 thiol-based redox mechanism in PKA repression.

## Discussion

Caloric restriction is established as a measure that extends the lifespan of organisms from yeast to primates and this effect occurs by reduced nutrient and/or growth signaling through the insulin/IGF-1, TOR and protein kinase A pathways. However, which effectors/processes downstream of these pathways are regulating the rate of aging is still a matter of controversy. As nutrient signaling coordinates many different cellular processes, the exact identity of the accountable process may differ between organisms and/or CR protocols (*Lamming and Anderson, 2014*). The fact that several of the target processes proposed, as for instance vacuolar pH control and protein homeostasis,

reciprocally feed-back control nutrient signaling (*Molin and Demir, 2014*; *Yao et al., 2015*; *Zhang et al., 2013*) has caused further obscured the designation of mechanisms important in slowing down aging. A novel integrative model of aging, however, posits that different pathways and/or organelles are intricately interconnected into so called integrons (*Dillin et al., 2014*), the interconnectivity of which eventually causes a progressive decline of all systems through sequential collapse of homeostasis, when individual subsystems fail.

Peroxiredoxins have emerged as regulators of aging stimulating longevity in organisms from yeast to worms, flies and mice (*De Haes et al., 2014*; *Hanzén et al., 2016*; *Lee et al., 2009*; *Molin et al., 2011*; *Oláhová and Veal, 2015*). We previously showed that the yeast peroxiredoxin Tsa1 is crucial for molecular chaperones to bind to aggregates forming in aged yeast cells (*Hanzén et al., 2016*), thus connecting peroxiredoxins to an aging factor conserved in many organisms. We linked this role to the sulfinylation of the enzyme primary catalytic cysteine and protein decamerization, thus providing a demonstration of the in vivo occurrence of this in vitro-described peroxiredoxin chaperone function (*Jang et al., 2004*; *Noichri et al., 2015*). We also previously observed that $H_2O_2$ resistance in CR cells requires both catalytic cysteines (*Molin et al., 2011*), and metformin, which extends lifespan in worms, causes the accumulation of disulfide-linked Prdx-2 in worms. These data indicated that handling protein aggregates might not be the only means by which peroxiredoxins regulate aging. Data reported in this study now demonstrate a key role of both cysteines of Tsa1 in slowing down aging, also correlating peroxiredoxin-stimulated longevity and hydrogen peroxide resistance. Surprisingly, the requirement for peroxiredoxin-catalytic cysteines in both aging and $H_2O_2$ resistance is not linked to $H_2O_2$ scavenging, but to the modulation of PKA. Taken together with the Tsa1-dependent increased lifespan in cells grown in the continuous presence of low levels of $H_2O_2$ (*Goulev et al., 2017*), these data demonstrate that at least two of the anti-aging effects of peroxiredoxins originates in $H_2O_2$ signaling. Accordingly, what are the phenotypes dependent on the scavenging function of Tsa1, and of peroxiredoxins in general? Compelling arguments for local scavenging by mouse PrdxI that modulate growth factor signaling have been made (*Woo et al., 2010*), but literature too often equate a requirement of peroxiredoxin catalytic cysteines with a role of the enzyme in scavenging. Our data now indicate that peroxiredoxins, when bearing its two catalytic residue, can override conventional second-messenger controlled signaling mechanisms to directly modulate protein kinase A signaling as a function of the level of $H_2O_2$ (*Figure 5J*).

How is this modulation of PKA by Tsa1 occuring? Our data provide evidence for a direct Tsa1-Tpk1 physical interaction, Tsa1-dependent Tpk1 cysteine sulfenylation and deglutathionylation, and a requirement of Cys243 in $H_2O_2$ resistance mediated by mild Tsa1 overexpression. Murine type II PKA is inactivated upon in vitro glutathionylation of the homologous Cys residue (C199) (*Humphries et al., 2005*; *Humphries et al., 2002*). In type II rat PKA, the same Cys residue forms a disulfide bond with the regulatory subunit at very low levels of $H_2O_2$ in vitro (1 µM), which decreases PKA activity (*de Piña et al., 2008*), again highlighting the importance of this residue in PKA redox regulation. The PKA regulatory subunit cysteine is however, only conserved in vertebrates, in contrast to the catalytic subunit cysteine, which is conserved in PKA across eukaryotes (*de Piña et al., 2008*). How does PKA then become redox modified? Are glutathionylation and sulfenylation of the PKA catalytic Cys residues, mechanistically linked, and if so which of them occurs first? Peroxiredoxins can oxidize other proteins by virtue of promiscuity, but disulfide bond formation and not sulfenylation is expected to occur in this case (*Stöcker et al., 2018b*). Furthermore, we could not identify a Tpk1-Tsa1 mixed disulfide by kinetic trapping using a Tsa1 mutant lacking its resolving cysteine (*Figure 4—figure supplement 1A–C*). Protein glutathionylation can occur non-enzymatically by condensation with a preformed sulfenate, a mechanism that may explain Tpk1 glutathionylation, but can also be catalyzed by a glutathione-S-transferase (*Zhang et al., 2018*). A pressing issue for the future will thus be to identify the mechanism by which Tpk1 becomes sulfenylated and glutathionylated and how peroxiredoxins, or possibly other redox enzymes assist these modifications.

The activities of both protein kinase G and A (PKARIα) are also stimulated by $H_2O_2$ (*Burgoyne et al., 2007*; *Burgoyne et al., 2015*). In the protein kinase G Iα isoform, a disulfide linking its two subunits forms in rat cells exposed to $H_2O_2$ (*Burgoyne et al., 2007*). Thus this regulation of PKA/PKG by $H_2O_2$ involves the same Cys195 conserved cysteine in the catalytic subunit but leads to opposite effects. Similarly, in vitro studies suggest that the energy-sensing kinase AMPK is activated upon glutathionylation (*Klaus et al., 2013*). In worms and mammals, the endoplasmic

reticulum (ER) transmembrane kinase Ire-1 is regulated by oxidation of another conserved Cys residue in the activation loop, situated 11 residues upstream of the here described PKA cysteine, at position +two relative to the $Mg^{2+}$-coordinating DFG motif (*Hourihan et al., 2016*). Furthermore, we recently found that another activation loop cysteine, positioned at DFG −1, in the fission yeast MAPKK, Wis1, restrains Wis1 activation by low levels, but not high, levels of $H_2O_2$ (*Sjölander et al., 2020*). These studies, together with the one presented here, pinpoint oxidation of cysteines in kinase activation loops as prevalent means of fine-tuning protein kinase function in response to $H_2O_2$.

In summary data presented here and in a previous study (*Hanzén et al., 2016*) point to two different independent mechanisms by which peroxiredoxins counteract aging and age-related disease (*Figure 6*). The first one, described here, involves catalytic cycling and inhibition of nutrient-related kinase signaling (*Figure 6A–B*). This mechanism appears critical for yeast to sustain normal longevity and is probably involved also in the ability of CR to slow down aging, since CR stimulates $H_2O_2$ resistance in a manner dependent on Tsa1 catalytic cysteines (*Molin et al., 2011*). Along the same lines, metformin-stimulated longevity in worms also seems to involve increased Prdx-2 disulfide bond formation (*De Haes et al., 2014*). The second mechanism is the stimulation of chaperone-dependent protein quality control that counteract protein aggregation (*Figure 6C*; *Hanzén et al., 2016*). Tsa1 sulfinylation is necessary to guide the molecular chaperones Hsp70 and Hsp104 to aggregates forming in aged and $H_2O_2$-treated cells. The requirement of both reduced PKA nutrient signaling and normal protein quality control (*Hanzén et al., 2016*) for mild Tsa1 overproduction to extend lifespan support a requirement of both these mechanisms for enhanced peroxiredoxin levels to extend lifespan (*Figure 6C*).

Cellular components and/or pathways that assimilate information from different subsystems, such as the above described nutrient signaling pathways, would thus be expected to have a key role as integrating hubs in the aging process. A role of PKA in integrating yeast homeostatic processes is also suggested by a genome-wide identification of genes controlling PKA regulatory-catalytic subunit interaction, and hence PKA activity, which found a striking number of known PKA targets, involved in glycogen accumulation, filamentous growth and amino-acid biosynthesis (*Filteau et al., 2015*). The role of peroxiredoxins of slowing down aging by modulating central nutrient signaling pathways agrees with the integrative model of aging and suggest that also other anti-aging regimens might impact nutrient signaling.

The incidence of many major age-related diseases, such as cancer, diabetes and neurodegeneration, can be reduced by caloric restriction (*Mattison et al., 2017*), and there is hope that reducing caloric intake or pharmaceutically targeting key molecular mechanisms underlying its beneficial health effects, such as peroxiredoxins, will fuel healthy, disease-free ageing. As peroxiredoxins are conserved in organisms from bacteria to humans and can be targeted pharmaceutically, they constitute promising targets for the development of drugs against age-related disease.

## Materials and methods

### Key resources table

| Reagent type (species) or resource | Designation | Source or reference | Identifiers | Additional information |
|---|---|---|---|---|
| Strain, strain background (*Escherichia coli*) | *E. coli* BL21 strain expressing pGEX2T-1-GST-RBD | This paper, 10.1038/s41467-017-01019-z | | To purify GST-RBD for Ras-GTP assays |
| Strain, strain background (*Saccharomyces cerevisiae*) | wt control | 10.1016/j.cell. 2016.05.006 | YMM130 | MAT alpha *his3Δ1:: pRS403, leu2Δ0 lys2Δ0 ura3Δ0* |
| Strain, strain background (*Saccharomyces cerevisiae*) | o/e *TSA1* | 10.1016/j.cell. 2016.05.006 | o/e *TSA1* | MAT alpha *his3Δ1:: pRS403-Myc-TSA1, leu2Δ0 lys2Δ0 ura3Δ0* |

*Continued on next page*

*Continued*

| Reagent type (species) or resource | Designation | Source or reference | Identifiers | Additional information |
|---|---|---|---|---|
| Strain, strain background (*Saccharomyces cerevisiae*) | *pde2Δ control* | This paper | YMM175 | MAT alpha *his3Δ1:: pRS403, leu2Δ0 lys2Δ0 ura3Δ0 pde2Δ::kanMX4* |
| Strain, strain background (*Saccharomyces cerevisiae*) | *pde2Δ o/e TSA1* | This paper | YMM176 | MAT alpha *his3Δ1:: pRS403-Myc-TSA1, leu2Δ0 lys2Δ0 ura3Δ0 pde2Δ:: kanMX4* |
| Strain, strain background (*Saccharomyces cerevisiae*) | wt | 10.1002/(SICI) 1097-0061(19980130)14: 2<115::AID-YEA204> 3.0.CO;2–2. | BY4742 | MAT alpha *his3Δ1 leu2Δ0 lys2Δ0 ura3Δ0* |
| Strain, strain background (*Saccharomyces cerevisiae*) | *tsa1Δ* | 10.1016/j.molcel.2011.07.027 | YMM114 | BY4742 *tsa1Δ::natMX4* |
| Strain, strain background (*Saccharomyces cerevisiae*) | *ras2Δ* | 10.1016/j.molcel.2011.07.027 | YMM113 | BY4742 *ras2Δ::kanMX4* |
| Strain, strain background (*Saccharomyces cerevisiae*) | *ras2Δtsa1Δ* | This paper | YMM170 | BY4742 *ras2Δ::kanMX4 tsa1Δ::natMX4* |
| Strain, strain background (*Saccharomyces cerevisiae*) | *pde2Δ* | Research Genetics, 10.1038/nature00935. | *pde2Δ* | BY4742 *pde2Δ::kanMX4* |
| Strain, strain background (*Saccharomyces cerevisiae*) | *ras2Δpde2Δ* | This paper | YMM171 | BY4742 *ras2Δ::kanMX4 pde2Δ::hphMX4* |
| Strain, strain background (*Saccharomyces cerevisiae*) | *pde2Δtsa1Δ* | This paper | YMM172 | BY4742 *pde2Δ::kanMX4 tsa1Δ::natMX4* |
| Strain, strain background (*Saccharomyces cerevisiae*) | *ras2Δpde2Δtsa1Δ* | This paper | YMM173 | BY4742 *ras2Δ::kanMX4 pde2Δ::hphMX4 tsa1Δ::natMX4* |
| Strain, strain background (*Saccharomyces cerevisiae*) | *tsa1C48S* | 10.1038/ncomms14791 | YMM145 | BY4742 *tsa1C48S* |
| Strain, strain background (*Saccharomyces cerevisiae*) | *tsa1C171S* | 10.1038/ncomms14791 | YMM146 | BY4742 *tsa1C171S* |
| Strain, strain background (*Saccharomyces cerevisiae*) | *tsa1ΔYF* | 10.1038/ncomms14791 | YMM147 | BY4742 *tsa1(1-184)* |
| Strain, strain background (*Saccharomyces cerevisiae*) | *tsa1C171SΔYF* | 10.1038/ncomms14791 | YMM148 | BY4742 *tsa1(1-184)C171S* |

*Continued*

| Reagent type (species) or resource | Designation | Source or reference | Identifiers | Additional information |
|---|---|---|---|---|
| Strain, strain background (*Saccharomyces cerevisiae*) | *trx1Δtrx2Δ* | 10.1038/ncomms14791 | YMM143 | BY4742 *trx1Δ::hphMX4 trx2Δ::natMX4* |
| Strain, strain background (*Saccharomyces cerevisiae*) | *msn2Δmsn4Δ* | This paper | YMM174 | BY4742 *msn2Δ::hphMX4 msn4Δ::natMX4* |
| Strain, strain background (*Saccharomyces cerevisiae*) | *ras1Δ::hphMX4* | This paper | YMM177 | MAT a, *his3Δ1 leu2Δ0 lys2Δ0 ura3Δ0 ras1Δ::hphMX4* |
| Strain, strain background (*Saccharomyces cerevisiae*) | | This paper | YMM178 | BY-2n *met15Δ0/MET15 lys2Δ0/LYS2 tpk1Δ::kanMX4/ TPK1 tpk2Δ::natMX4/ TPK2 tpk3Δ:: hphMX4/TPK3* |
| Strain, strain background (*Saccharomyces cerevisiae*) | *tpk1Δtpk3Δ* | This paper | YMM179 | BY4742 *tpk1Δ::kanMX4 tpk3Δ::hphMX4* |
| Strain, strain background (*Saccharomyces cerevisiae*) | *tpk2Δtpk3Δ* | This paper | YMM180 | BY4742 *tpk2Δ::natMX4 tpk3Δ::hphMX4* |
| Strain, strain background (*Saccharomyces cerevisiae*) | *tpk1Δtpk2Δ tpk3Δ pTPK1-URA* | This paper | YMM181 | BY4742 *tpk1Δ::kanMX4 tpk2Δ::natMX4 tpk3Δ::hphMX4 pRS316-TPK1* |
| Strain, strain background (*Saccharomyces cerevisiae*) | *tpk1Δtpk2Δtpk3Δ pTPK1-URA vector control* | This paper | YMM182 | BY4742 *tpk1Δ::kanMX4 tpk2Δ::natMX4 tpk3Δ::hphMX4 pRS313 pTPK1-URA3* |
| Strain, strain background (*Saccharomyces cerevisiae*) | *tpk1Δtpk2Δtpk3Δ pTPK1-URA pTPK1* | This paper | YMM183 | BY4742 *tpk1Δ::kanMX4 tpk2Δ::natMX4 tpk3Δ::hphMX4 pRS313-TPK1 pTPK1-URA3* |
| Strain, strain background (*Saccharomyces cerevisiae*) | *tpk1Δtpk2Δtpk3Δ pTPK1-URA3 ptpk1C243A* | This paper | YMM184 | BY4742 *tpk1Δ::kanMX4 tpk2Δ::natMX4 tpk3Δ::hphMX4 pRS313-tpk1C243A pTPK1-URA3* |
| Strain, strain background (*Saccharomyces cerevisiae*) | *tpk1Δtpk2Δtpk3Δ pTPK1-URA3 ptpk1C243D* | This paper | YMM185 | BY4742 *tpk1Δ::kanMX4 tpk2Δ::natMX4 tpk3Δ::hphMX4 pRS313-tpk1C243D pTPK1-URA3* |
| Strain, strain background (*Saccharomyces cerevisiae*) | *tpk1Δtpk2Δtpk3Δ pTPK1-URA3 ptpk1T241A* | This paper | YMM186 | BY4742 *tpk1Δ::kanMX4 tpk2Δ::natMX4 tpk3Δ::hphMX4 pRS313-tpk1T241A pTPK1-URA3* |

*Continued on next page*

Continued

| Reagent type (species) or resource | Designation | Source or reference | Identifiers | Additional information |
|---|---|---|---|---|
| Strain, strain background (*Saccharomyces cerevisiae*) | *tpk1Δtpk2Δtpk3Δ pTPK1* | This paper | YMM187 | BY4742 *tpk1Δ::kanMX4 tpk2Δ::natMX4 tpk3Δ::hphMX4 pRS313-TPK1* |
| Strain, strain background (*Saccharomyces cerevisiae*) | *tpk1Δtpk2Δtpk3Δ ptpk1C243A* | This paper | YMM188 | BY4742 *tpk1Δ::kanMX4 tpk2Δ::natMX4 tpk3Δ::hphMX4 pRS313-tpk1C243A* |
| Strain, strain background (*Saccharomyces cerevisiae*) | *tpk1Δtpk2Δtpk3Δ ptpk1C243D* | This paper | YMM189 | BY4742 *tpk1Δ::kanMX4 tpk2Δ::natMX4 tpk3Δ::hphMX4 pRS313-tpk1C243D* |
| Strain, strain background (*Saccharomyces cerevisiae*) | *tpk1Δtpk2Δtpk3Δ ptpk1T241A* | This paper | YMM190 | BY4742 *tpk1Δ::kanMX4 tpk2Δ::natMX4 tpk3Δ::hphMX4 pRS313-tpk1T241A* |
| Strain, strain background (*Saccharomyces cerevisiae*) | *ras2Δtrx1Δtrx2Δ* | This paper | YMM191 | BY4742 *ras2Δ::kanMX4 trx1Δ::hphMX4 trx2Δ::natMX4* |
| Strain, strain background (*Saccharomyces cerevisiae*) | *tsa1Δ::bleMX4* | This paper | YMM192 | BY4741 *tsa1Δ::bleMX4* |
| Strain, strain background (*Saccharomyces cerevisiae*) | *tpk2Δtpk3Δtsa1Δ* | This paper | YMM193 | BY4741 *tpk2Δ::natMX4 tpk3Δ::hphMX4 tsa1Δ::bleMX4* |
| Strain, strain background (*Saccharomyces cerevisiae*) | *TPK1-HBH tpk2Δtpk3Δ* | This paper | WR1832 | BY4742 *TPK1-HBH::TRP1 tpk2Δ::natMX4 tpk3Δ::hphMX4 trp1Δ::kanMX4* |
| Strain, strain background (*Saccharomyces cerevisiae*) | *tpk1Δtpk2Δtpk3Δ pTPK1-URA vector control* | This paper | yCP101 | MAT a *his3Δ1::pRS403, leu2Δ0 lys2Δ0 ura3Δ0 tpk1Δ::kanMX4 tpk2Δ::natMX4 tpk3Δ::hphMX4 pRS316-TPK1* |
| Strain, strain background (*Saccharomyces cerevisiae*) | *tpk1Δtpk2Δtpk3Δ ptpk1C243A-URA vector control* | This paper | yCP102 | MAT alpha *his3Δ1:: pRS403, leu2Δ0 lys2Δ0 ura3Δ0 tpk1Δ::kanMX4 tpk2Δ::natMX4 tpk3Δ::hphMX4 pRS316-tpk1C243A* |
| Strain, strain background (*Saccharomyces cerevisiae*) | *tpk1Δtpk2Δtpk3Δ pTPK1-URA o/e TSA1* | This paper | yCP103 | MAT alpha *his3Δ1:: pRS403-myc-TSA1, leu2Δ0 lys2Δ0 ura3Δ0 tpk1Δ::kanMX4 tpk2Δ::natMX4 tpk3Δ::hphMX4 pRS316-TPK1* |

*Continued on next page*

*Continued*

| Reagent type (species) or resource | Designation | Source or reference | Identifiers | Additional information |
|---|---|---|---|---|
| Strain, strain background (*Saccharomyces cerevisiae*) | *tpk1Δtpk2Δtpk3Δ ptpk1C243A-URA o/e TSA1* | This paper | yCP104 | MAT alpha *his3Δ1:: pRS403-myc-TSA1, leu2Δ0 lys2Δ0 ura3Δ0 tpk1Δ::kanMX4 tpk2Δ::natMX4 tpk3Δ::hphMX4 pRS316-tpk1C243A* |
| Strain, strain background (*Saccharomyces cerevisiae*) | *tpk1Δtpk2Δtp k3Δtsa1Δ pTPK1* | This paper | yCP105 | BY4742 *tpk1Δ::kanMX4 tpk2Δ::natMX4 tpk3Δ::hphMX4 tsa1Δ::bleMX4 pRS313-TPK1* |
| Strain, strain background (*Saccharomyces cerevisiae*) | *tpk1Δtpk2Δ tpk3Δtsa1Δ ptpk1T241A* | This paper | yCP106 | BY4742 *tpk1Δ::kanMX4 tpk2Δ::natMX4 tpk3Δ::hphMX4 tsa1Δ::bleMX4 pRS313-tpk1T241A* |
| Strain, strain background (*Saccharomyces cerevisiae*) | *TPK1-HBH tpk 2Δtpk3Δtsa1Δ* | This paper | yCP107 | BY4742 *TPK1-HBH:: TRP1 tpk2Δ::natMX4 tpk3Δ::hphMX4 tsa1Δ::bleMX4 trp1Δ::kanMX4 tsa1Δ::bleMX4* |
| Antibody | (mouse monoclonal) anti-Tpk1 | Santa Cruz Biotechnology | Sc-374592, RRID:AB_10990730 | (1:1000) |
| Antibody | (goat polyclonal) anti-Bcy1 | Santa Cruz Biotechnology | Sc-6734, RRID:AB_671758 | (1:2000) |
| Antibody | (rabbit) IgG; anti-Protein A | Sigma Aldrich | I5006, RRID:AB_1163659 | 1 µg/ml |
| Antibody | (goat polyclonal) anti-Ras2 | Santa Cruz Biotechnology | Sc-6759, RRID:AB_672465 | (1:2000) |
| Antibody | (mouse monoclonal) anti-Glutathione (D8) | Abcam | ab19534, RRID:AB_880243 | (1:1000) |
| Antibody | (mouse monoclonal) anti-Pgk1 (22C5D8) | Thermo Fisher | 459250, RRID:AB_2532235 | (1:500) |
| Antibody | (mouse monoclonal) anti-2 Cys Prx (6E5); (anti-Tsa1) | Abcam | ab16765, RRID:AB_443456 | (1:1000) |
| Recombinant DNA reagent | yEP24 | 10.1016/0378-1119(79)90004-0 | | yeast 2µ, *URA3* vector plasmid |
| Recombinant DNA reagent | pKF56 | 10.1128/mcb. 10.8.4303. | | *IRA2* in yEP24 |
| Recombinant DNA reagent | pRS425 | 10.1016/0378-1119(92)90454w. | | yeast 2µ, *LEU2* vector plasmid |
| Recombinant DNA reagent | yEP13-*PDE2* | 10.1093/emboj/cdg314. | | *PDE2* in yeast 2µ, *LEU2* plasmid |
| Recombinant DNA reagent | yEPlac195 | 10.1016/0378-1119(88)90185-0. | | yeast 2µ, *URA3* vector plasmid |
| Recombinant DNA reagent | pXP1 | 10.1128/mcb.19.7.4874. | | *BCY1* in yEPlac195 |
| Recombinant DNA reagent | pRS315 | PMID:2659436 | | yeast CEN/ARS, *LEU2* empty vector plasmid |
| Recombinant DNA reagent | B561 (pRS315-*RAS2G19V*) | 10.1128/mcb. 19.10.6775. | | *RAS2G19V* in pRS315 |

*Continued on next page*

*Continued*

| Reagent type (species) or resource | Designation | Source or reference | Identifiers | Additional information |
|---|---|---|---|---|
| Recombinant DNA reagent | pHyPer3C199S (pRS416-*GPD*-HyPer3C199S) | This paper, 10.1021/cb300625g | | HyPer3C199S |
| Recombinant DNA reagent | pRS416-*GPD-AKAR4* | *Molin et al., 2020* | | AKAR4 in pRS416-GPD [CEN/ARS, pGPD promotor, *URA3*] |
| Recombinant DNA reagent | pRS316 | PMID:2659436 | | yeast CEN/ARS, *URA3* empty vector plasmid |
| Recombinant DNA reagent | pRS316- *myc-TSA1* | 10.1038/nature02075. | | Myc-*TSA1* in pRS316 |
| Recombinant DNA reagent | pRS316- *myc-tsa1C48S* | 10.1016/j.molcel.2011.07.027 | | Myc-*tsa1C48S* in pRS316 |
| Recombinant DNA reagent | pRS316- *myc-tsa1C171S* | 10.1016/j.molcel.2011.07.027 | | Myc-*tsa1C171S* in pRS316 |
| Recombinant DNA reagent | pRS315-ProtA | This paper | | ProteinA in pRS315 |
| Recombinant DNA reagent | pRS315-*TRX2-ProteinA* | 10.1038/ncomms14791 | | *TRX2-ProtA* in pRS315 |
| Recombinant DNA reagent | pRS315-*trx2C 34S-ProteinA* | This paper | | *trx2C34S-ProtA* in pRS315 |
| Recombinant DNA reagent | pRS315-*trx2C31 SC34S-ProteinA* | This paper | *trx2C31SC34S-ProtA* in pRS315 | *trx2C31SC34S-ProtA* in pRS315 |
| Recombinant DNA reagent | pRS313 | PMID:2659436 | yeast CEN/ARS, *HIS3* empty vector | yeast CEN/ARS, *HIS3* empty vector |
| Recombinant DNA reagent | pRS313-*TPK1* | 10.1074/jbc.M110.200071. | *TPK1* in pRS313 | *TPK1* in pRS313 |
| Recombinant DNA reagent | pRS313-*tpk1C243A* | This paper | *tpk1C243A* in pRS313 | *tpk1C243A* in pRS313 |
| Recombinant DNA reagent | pRS313-*tpk1C243D* | This paper | *tpk1C243D* in pRS313 | *tpk1C243D* in pRS313 |
| Recombinant DNA reagent | pRS313-*tpk1T241A* | This paper | *tpk1T241A* in pRS313 | *tpk1T241A* in pRS313 |
| Recombinant DNA reagent | p*TPK1-URA3* (pRS316-*TPK1*) | Karin Voordeckers | *TPK1* in pRS316 | *TPK1* in pRS316 |
| Recombinant DNA reagent | p*tpk1C243A-URA3* | This paper | *tpk1C243A* in pRS316 | *tpk1C243A* in pRS316 |
| Sequence-based reagent | *ACT1F* | 10.1016/j.molcel.2011.03.021 | For Q-PCR of *ACT1* | CTGCCGGTATTGACCAAACT |
| Sequence-based reagent | *ACT1R* | 10.1016/j.molcel.2011.03.021 | For Q-PCR of *ACT1* | CGGTGAATTTCCTTTTGCATT |
| Sequence-based reagent | *CTT1F* | This paper | For Q-PCR of *CTT1* | GCTTCTCAATACTCAAGACCAG |
| Sequence-based reagent | *CTT1R* | This paper | For Q-PCR of *CTT1* | GCGGCGTATGTAATATCACTC |
| Sequence-based reagent | *HSP12F* | 10.1016/j.molcel.2011.03.021 | For Q-PCR of *HSP12* | AGGTCGCTGGTAAGGTTC |
| Sequence-based reagent | *HSP12R* | 10.1016/j.molcel.2011.03.021 | For Q-PCR of *HSP12* | ATCGTTCAACTTGGACTTGG |
| Peptide, recombinant protein | Glutathione-S-Transferase-Raf1-Binding-Domain (GST-RBD) | This paper, 10.1038/s41467-017-01019-z | For Ras-GTP assay | Purified from *E. coli* strain BL21 expressing pGEX2T-1-GST-RBD |

*Continued on next page*

*Continued*

| Reagent type (species) or resource | Designation | Source or reference | Identifiers | Additional information |
|---|---|---|---|---|
| Commercial assay or kit | PureLink RNA Mini kit | Thermo-Fisher | Cat #: 12183025 | |
| Commercial assay or kit | QuantiTect Reverse Transcription Kit | Qiagen | Cat #: 205313 | |
| Commercial assay or kit | iQ SYBR Green Supermix | BioRad | Cat #: 170–8882 | |
| Commercial assay or kit | LANCE cAMP 384 kit | Perkin Elmer | Cat #: AD0262 | |
| Chemical compound, drug | G418 | Acros Organics | Cat #: 329400050 | |
| Chemical compound, drug | ClonNAT | Werner Bioagents | Cat #: 5.005.000 | |
| Chemical compound, drug | Hygromycin B | Formedium | Cat #: HYG5000 | |
| Chemical compound, drug | Phleomycin | Sigma Aldrich | P9564 | |
| Chemical compound, drug | 5-fluoroorotic acid | Sigma Aldrich | F5013 | |
| Chemical compound, drug | EZ-Link Sulfo-NHS-LC Biotin | Thermo Fisher | Cat #: 21335 | |
| Chemical compound, drug | Trichloroacetic acid | Sigma Aldrich | Cat #: T6399 | |
| Chemical compound, drug | KSCN | Sigma Aldrich | Cat #: P2713 | |
| Chemical compound, drug | $(NH_4)_2Fe(SO_4)_2 \cdot 6 H2O$ | Sigma Aldrich | Cat #: 215406 | |
| Chemical compound, drug | TRIzol Reagent | Thermo Fisher | Cat #: 15596026 | |
| Chemical compound, drug | DNase, RNase-free set | Qiagen | Cat #: 79254 | |
| Chemical compound, drug | cOmplete Mini EDTA-free protease inhibitor | Roche Applied Science | Cat #: 11873580001 | |
| Chemical compound, drug | Glutathione Sepharose beads | GE Healthcare | Cat #: 17-0756-01 | |
| Chemical compound, drug | 12% Bis-Tris NUPAGE gels | Thermo FisherArch Biochem Biophys | Cat #: NP0349BOX | |
| Chemical compound, drug | MOPS running buffer | Thermo Fisher | Cat #: NP0001 | |
| Chemical compound, drug | Immobilon-FL PVDF membrane | Millipore | Cat #: IPFL00010 | |
| Chemical compound, drug | $Ni^{2+}$-Sepharose beads | GE Healthcare | Cat #: 17-5318-06 | |
| Chemical compound, drug | Anti-c-myc, agarose conjugated | Sigma-Aldrich | Cat #: A7470 | |
| Chemical compound, drug | Trypsin Gold, mass spectrometry grade | Promega | Cat #: V5280 | |
| Chemical compound, drug | N-ethylmaleimide | Sigma-Aldrich | Cat #: E3876 | |
| Chemical compound, drug | DYn-2 | Cayman Chemical | Cat #: 11220 | |

*Continued*

| Reagent type (species) or resource | Designation | Source or reference | Identifiers | Additional information |
|---|---|---|---|---|
| Chemical compound, drug | 10% Criterion TGX Precast Midi Protein Gel | Bio-Rad | Cat #: 5671034 | |
| Chemical compound, drug | Peptide Retention Time Calibration Mixture | Pierce, Thermo Fisher | Cat #: 88320 | |
| Software, algorithm | MATLAB | Mathworks | version 2016b | |
| Software, algorithm | CellX | 10.1002/0471142727. mb1422s101 | | |
| Software, algorithm | Scrödinger Suite | Schrödinger LLC | | |
| Software, algorithm | GROMACS | 10.1016/j. softx.2015.06.001 | | |
| Software, algorithm | Amber tools | 10.1002/wcms.1121 | | |

## Strains and growth conditions

Yeast strains and plasmids are listed in the Key resources table. The strains used in this study are derivatives of BY4741/BY4742. Strains were grown at 30°C in YPD 2% glucose (w/v) or in Yeast Nitrogen Base defined medium containing 2% glucose and complete supplement mixture (CSM) lacking the appropriate amino acids (Formedium) as described previously (*Molin et al., 2011*). To check the segregation of deletion markers in tetrad dissections YPD medium supplied with the following chemicals was used to check segregation of the dominant markers: *kanMX4* (G418 200 µg/ml), *natMX4* (ClonNAT 100 µg/ml), *hphMX4* (Hygromycin B 300 µg/ml), *bleMX4* (Phleomycin 40 µg/ml). To counterselect the p*TPK1-URA3* plasmid cells were grown in defined glucose CSM –HIS, 5-FOA medium containing YNB, glucose and CSM –URA, HIS; 50 mg/l uracil and 1 g/l 5-fluoroorotic acid.

## Strain and plasmid constructions

Strains YMM170 (*ras2Δtsa1Δ*) and YMM172 (*pde2Δtsa1Δ*) were constructed by crossing strain YMM114 to BY4741 *ras2::kanMX4* and BY4741 *pde2Δ::kanMX4* (Research Genetics, *Giaever et al., 2002*), respectively, and selecting for Mat alpha, methionine prototrophic, lysine auxotrophic, G418 resistant and nourseothricin resistant progeny. Strains YMM171 and YMM173 were constructed by crossing a BY4741 *pde2Δ::hphMX4* {*pde2Δ::kanMX4* from the deletion collection (Research Genetics, *Giaever et al., 2002*) marker-switched (*Goldstein and McCusker, 1999*) to *pde2Δ::hphMX4* to strain YMM170 (*ras2Δtsa1Δ*) and selecting for Mat alpha, methionine prototrophic, lysine auxotrophic, G418 resistant, hygromycin resistant and nourseothricin sensitive (YMM171 *ras2Δpde2Δ*) or nourseothricin resistant (YMM173 *ras2Δpde2Δtsa1Δ*) progeny. Strains YMM174 (*msn2Δmsn4Δ*), YMM175 (*pde2Δ*) and YMM176 (*pde2Δ* o/e *TSA1*) were constructed by crossing BY4741 *msn2Δmsn4Δ* (*Caballero et al., 2011*) or BY4741 *pde2Δ::kanMX4* (Research Genetics, *Giaever et al., 2002*) to strains YMM130 or BY4742 *his3Δ1::pRS403-Myc-TSA1*, respectively and selecting for Mat alpha, methionine prototrophic, lysine auxotrophic, histidine auxotrophic, hygromycin- and nourseothricin-resistant progeny (YMM174) or Mat alpha, methionine prototrophic, lysine auxotrophic, G418 resistant and histidine prototrophic progeny (YMM174 and YMM175). Strain YMM177 was constructed by marker-switching (*Goldstein and McCusker, 1999*) a Mat a *ras1Δ::kanMX4* spore, obtained from crossing strain BY4741 *ras1Δ::kanMX4* Research Genetics, *Giaever et al., 2002*) to strain YMM114 and selecting for Mat a, methionine prototrophic, lysine auxotrophic and G418 resistant progeny, to *ras1Δ::hphMX4*. Strain YMM178 (*tpk1Δ/TPK1 tpk2Δ/TPK2 tpk3Δ/TPK3*) was constructed by crossing a BY4742 *tpk1Δ::kanMX4 tpk2Δ::natMX4* strain to a BY4741 *tpk3Δ::hphMX4* strain {*tpk3Δ::kanMX4* from the deletion collection [Research Genetics, (*Giaever et al., 2002*) marker-switched (*Goldstein and McCusker, 1999*) to *tpk3Δ::hphMX4* resulting in a *tpk1Δ/TPK1 tpk2Δ/TPK2 tpk3Δ/TPK3* heterozygous diploid strain. A Mat alpha, G418- and

hygromycin-resistant spore constitutes strain YMM179 whereas a Mat alpha, nourseothricin- and hygromycin-resistant spore constitutes strain YMM180. The BY4742 *tpk1Δ::kanMX4 tpk2Δ::natMX4* strain was constructed by introducing *tpk2Δ::natMX4* PCR amplified from a BY4742 *tpk2Δ::natMX4* strain (*Costanzo et al., 2010*) into strain BY4741 *tpk1Δ::kanMX4* [Research Genetics, (*Giaever et al., 2002*) selecting for nourseothricin- and G418-resistance and verifying the deletion by diagnostic PCR. A BY4742 *tpk1Δtpk2Δtpk3Δ* p*TPK1-URA3* haploid strain (YMM181) was constructed by transforming strain YMM177 with plasmid p*TPK1-URA3* and sporulating the strain selecting for a Mat alpha methionine prototrophic, lysine auxotrophic, G418-, nourseothricin-, hygromycin B-resistant and uracil auxotrophic progeny. Strains YMM182-YMM186 were constructed by transforming strain YMM180 with plasmids pRS313 (YMM181), pRS313-*TPK1* (YMM183), pRS313-*tpk1C243A* (YMM184) and pRS313-*tpk1C243D* (YMM185) and pRS313-*tpk1T241A* (YMM186). Counterselecting p*TPK1-URA3* on 5-FOA medium resulted in strains YMM187 (BY4742 *tpk1Δtpk2Δtpk3Δ* pRS313-*TPK1*), YMM188 (BY4742 *tpk1Δtpk2Δtpk3Δ* pRS313-*tpk1C243A*), YMM189 (BY4742 *tpk1Δtpk2Δtpk3Δ* pRS313-*tpk1C243D*) and yMM190 (BY4742 *tpk1Δtpk2Δtpk3Δ* pRS313-*tpk1T241A*), respectively. Strain YMM191 (*ras2Δtrx1Δtrx2Δ*) was constructed by crossing strain YMM113 (*ras2Δ*) to strain YMM143 (*trx1Δtrx2Δ*) selecting for Mat alpha, methionine prototrophic, lysine auxotrophic, G418-, nourseothricin- and hygromycin B-resistant progeny. Strain YMM192 was constructed by marker-switching strain BY4741 *tsa1Δ::kanMX4* [Research Genetics, (*Giaever et al., 2002*) into BY4741 *tsa1Δ::bleMX4* using a bleMX4 cassette PCR amplified from plasmid pUG66 (*Gueldener et al., 2002*) using primers PR78 and PR79 (*Goldstein and McCusker, 1999*). Strain yMM193 was constructed by crossing strains yMM180 and yMM192 selecting for a Mat a, nourseothricin+, hygromycin+ and phleomycin+ spore. Strain WR1832 was constructed by first introducing PCR amplified *trp1Δ::kanMX4* DNA (*Longtine et al., 1998*) into strain YMM180, verification of cassette integration by PCR and loss of the ability to grow without tryptophan supplement and next by *HBH::TRP1* C-terminal tagging of *TPK1* and PCR based verification as described (*Tagwerker et al., 2006*). Strains yCP101-yCP104 were constructed by crossing Mat a *his3Δ1::pRS403* or *his3Δ1::pRS403-myc-TSA1* spores, obtained in crosses generating strains yMM175 above, either to strain yMM183 or to strain yMM187 also carrying plasmid *pRS316-tpk1C243A*. Methionine prototrophic, lysine auxotrophic, histidine prototrophic, 5-FOA-sensitive, G418+, nourseothricin+ and hygromycin B+ progeny obtained in these crosses constitute strains yCP101-yCP104 listed in *Supplementary file 1* Table S1. Strains yCP105 and yCP106 were constructed by crossing strains yMM187 (p*TPK1*) or yMM189 (p*tpk1T241A*), respectively, to strain yMM192 selecting for Mat alpha, Met+, Lys-, G418+, Nat+, Hyg+, Phleomycin+, His+ progeny. Strain yCP107 was constructed by crossing strain WR1832 to yMM193 and selecting for Mat alpha, Met+, Lys-, G418+, Nat+, Hyg+, Phleomycin+, Trp+ progeny.

Plasmids pRS313-*tpk1C243A*, pRS313-*tpk1C243D*, pRS313-*tpk1T241A* and pRS316-*tpk1C243A*, were constructed by site directed mutagenesis of the pRS313-*TPK1* or pRS316-*TPK1* plasmids (Eurofins Genomics). Plasmids pRS315-*trx1C34S*-ProtA and pRS315-*trx2C31SC34S*-ProtA were constructed by site-directed mutagenesis of plasmid pRS315-*TRX2*-ProtA (GenScript). The correct sequence of all plasmids constructed was verified by sequencing.

## Lifespan analyses

Lifespan analyses were performed as previously described by counting the number of daughters produced in a cohort of mother cells (*Erjavec et al., 2007*).

## 2D-page

Protein synthesis rates of the indicated proteins were determined in $^{35}$S-Methionine labelled protein extracts separated by two-dimensional polyacrylamide gel electrophoresis as described (*Maillet et al., 1996*; *Molin et al., 2011*). Tsa1 sulfinylation was determined by comparing levels of sulfinylated Tsa1 (Tsa1-SOOH) to non-sulfinylated Tsa1 on silver-stained 2D gels as described (*Molin et al., 2011*).

## Spot tests

$H_2O_2$ resistance was tested with mid-exponential-phase ($A_{600}$ = 0.3, $3 \times 10^6$ cells/ml) cells that were diluted (x5, x50, x500, x5000, x50000) and spotted onto SD media containing 0 to 1 mM $H_2O_2$ or

YPD media containing 0 to 2 mM. The number of colonies after 2 days incubation at 30 ∘C on $H_2O_2$ plates was divided with the number on control plates to get $H_2O_2$-resistance (%).

For glycogen accumulation, plates incubated for 2 days at 30 ∘C were exposed to iodine-bead fumes for 2.5 min and scanned immediately.

## Spore viability

The viability of spores segregating in the sporulation and dissection of a heterozygous diploid *ras1Δ:: hphMX4/RAS1 ras2Δ::kanMX4/RAS2 tsa1Δ::natMX4/TSA1* strain obtained by crossing strain YMM176 (*ras1Δ::hphMX4*) to strain YMM170 (BY4742 *ras2Δ::kanMX4 tsa1Δ::natMX4*) was analyzed after 4 days of incubation at 30°C in 32 tetrads where 1) all markers analyzed (*hphMX4, kanMX4, natMX4, MET15, LYS2*) segregated 2:2, 2) the exact genotypes of all spores were possible to deduce from this information and 3) the genotypes of dead spores were assigned based on markers present in the other spores dissected from the same tetrads. Similarly, spore viability of spores segregating in a heterozygous diploid *tpk1Δ::kanMX4/TPK1 tpk2Δ::natMX4/TPK2 tpk3Δ::hphMX4/TPK3 tsa1Δ:: bleMX4/TSA1*, obtained by crossing strain YMM191 (BY4741 *tsa1Δ::bleMX4*) to strain YMM186 (BY4742 *tpk1Δ::kanMX4 tpk2Δ::natMX4 tpk3Δ::hphMX4* expressing pRS313-*TPK1*), was analyzed in 43 tetrads where all chromosomal markers analyzed (*kanMX4, natMX4, hphMX4, bleMX4, MET15, LYS2*) segregated 2:2. The ability to grow in the absence of histidine supplementation (-HIS) was taken as an indication that the pRS313-*TPK1* plasmid was present.

## Quantitative Real-Time PCR analysis

Cell cultures were harvested in mid-exponential phase and resuspended in 1 ml Trizol Reagent (Invitrogen) and homogenized with silica beads by Fast prep (6.5 m/s, 30 s, interval 2.5 min, 4 ℃). RNA was extracted using phenol chloroform extraction and precipitated with sodium acetate/ethanol. The pellet was treated with DNase for 30 min followed by heat-inactivation of the enzyme. The RNA was purified with Invitrogen PureLink RNA Mini Kit columns and converted to cDNA following the QIAGEN QuantiTect Reverse Transcription Kit. Q-PCR was performed with 50 ng cDNA by using BioRad iQ SYBR Green Supermix and quantified with the BioRad iCycler, iQ5. Relative levels of mRNA were calculated by using cycle times of *ACT1* as a reference gene.

## Quantitative analyses of Msn2-GFP localization

Msn2-signaling was analyzed as described previously (*Bodvard et al., 2017*). Briefly, the fraction of cells displaying nuclear localization of Msn2-GFP (nucleus/cytoplasm signal ratio >1.28) at each time point was calculated and used to calculate the total time Msn2 spent in the nucleus during a 60 min experiment.

## Measurement of Ras2-GTP in vivo

Ras2-GTP level was measured as a ratio between Ras2-GTP and total Ras2 as described previously (*Colombo and Martegani, 2014*; *Peeters et al., 2017*). Mid-exponential phase yeast cells were harvested and lysed with glass- beads in Fast-prep (6.0 m/s, 20 s, interval 2.5 min) in lysis buffer [50 mM Tris-HCl, 200 mM NaCl, 2.5 mM $MgCl_2$, 10% glycerol, 1% Triton X100, cOmplete Protease inhibitor EDTA-free]. The supernatant with 1.5 mg of total protein was incubated with a bed volume 50 μL of glutathione S-transferase (GST)-RBD fusion protein pre-bound to glutathione-Sepharose for 1 hr at 4 ℃ and washed three times with lysis buffer by centrifugation. For elution the beads were boiled for 5 min at 98 ℃ in SDS-sample buffer (6% SDS, 62.5 mM Tris-HCl pH 8.7, 30% Glycerol, 0.75% β-mercaptoethanol). Through western blotting, Ras2-GTP and total Ras2 proteins were detected with anti-Ras2 antibodies. Determination of ratios between Ras2-GTP and total Ras2 was performed by ImageJ.

cAMP measurement cAMP measurements were performed as previously described (*Caballero et al., 2011*; *Parts et al., 2011*). $2 \times 10^8$ cells grown to midexponential phase were pelleted, washed, and resuspended in 1 ml cold milliQ water. Metabolites were extracted by adding 1.2 ml TCA (0.5 M) and occasional vigourous vortexing while samples were kept on ice for 15 min. TCA was removed by ether extraction. cAMP levels were determined by the LANCE cAMP 384 kit in 40 μL total reactions and by comparing to the standards supplied. The values for cAMP were normalized to the wild type level.

## Global H$_2$O$_2$ scavenging in the medium

Medium peroxide determinations were performed using a ferrithiocyanate spectrophotometric assay (*Molin et al., 2007*). After bolus addition of H$_2$O$_2$, 100 µL sample aliquots were withdrawn and cultures were arrested by the addition of 1 ml ice-cold 10% TCA. After pelleting cells 180 mM KSCN and 1.4 mM Fe(NH$_4$)$_2$(SO$_4$)$_2$ final concentrations were added to the supernatants. Absorbance at 480 nm was subsequently determined and compared to equally TCA-treated H$_2$O$_2$ standards diluted in medium.

## Isolation of old cells

Old cells were obtained as previously described by sorting biotin-labeled mother cells using the MagnaBind streptavidin system (*Sinclair and Guarente, 1997*). Briefly, mid-exponential phase cells were labeled with EZ-Link Sulfo-NHS-LC Biotin and grown overnight in minimal media (CSM-His). The cells were incubated with streptavidin-conjugated magnetic beads for 2 hr and then sorted magnetically with the unlabeled cells being washed away. Sorted cells were then grown overnight and the streptavidin labeling procedure was repeated before sorting one last time. After sorting the cells were incubated for 1 hr in CSM-His media at 30 ∘C for recovery before microscopy.

## Measurements of cytoplasmic H$_2$O$_2$ using HyPer3

Fluorescence of the ratiometric probe HyPer-3 (*Bilan et al., 2013*) was acquired using an Olympus IX81 motorized microscope with a PlanApoN 60x/1.42 Oil objective and a 12-bit Hamamatsu camera. Shifts in the fluorescence intensities were acquired with excitation around 500 nm (485/20 nm) and 420 nm (427/10 nm filter) and an emission filter around 520 nm (Fura two filter). For bolus addition of H$_2$O$_2$, cells in midexponential phase were incubated with 0.2 mM H$_2$O$_2$ for 10 min and immediately imaged.

## Image analysis of HyPer3 fluorescence

Image and signal analysis was performed using the MATLAB toolbox 2016b. Cell segmentation is performed with the CellX algorithm using the bright-field channel. The fluorescent intensity data were obtained from fluorescent images and data are presented as the median 500 nm fluorescent signal normalized to the median fluorescent 420 nm signal by dividing the latter with the former.

## AKAR4 FRET-based PKA activity measurements

Detection of cyan fluorescent protein CFP to yellow fluorescent protein YFP FRET in the AKAR4 sensor was performed as described previously (*Depry and Zhang, 2011*; *Molin et al., 2020*). CFP was excited at 427/10 nm, YFP was excited at 504/6 nm and emission was monitored using a Semrock dual bandpass filter (part no: FF01-464/547). Images were acquired using an automated epi-fluorescence microscope (Olympus IX81) equipped with a × 60 oil-immersion objective (numerical aperture 1.4, PlanApoN ×60/1.42 Oil, Olympus) and an electron-multiplying charge-coupled device camera (12-bit Hamamatsu camera). The yeast cells were kept in a heated perfusion chamber (FCS2, Bioptechs Inc) at 28℃ to avoid heat-induced stress responses. The objective was heated to 26.2℃ (according to the manufacturer's instructions) to maintain a stable temperature in the perfusion chamber. The cover glasses were precoated for 1.5 hr with protein concanavalin A, 0.5 µg µl−1 in 0.01 M PBS, to immobilize yeast cells on the surface.

## Immunoprecipitation

Cells from 50 mL/sample of mid-exponential phase YPD culture was pelleted, the pellet was washed with cold water and pelleted again, washed with 1 mL lysis buffer (50 mM Tris HCl pH 8.0, 150 mM NaCl, 1 mM EDTA, 10% Glycerol, 5 mM MgCl$_2$ and protease-inhibitor cocktail). Cells were broken in 0.35 mL lysis buffer by beads at four degrees in a Fastprep FP120 cell disrupter (Bio101/ThermoSavant, speed 5 m/sec, 4 times 40 s with >1 min on ice in between each agitation). The extract was pelleted at 12500 rpm at four degrees and the supernatant was used for subsequent analyses. An aliquot of supernatant was withdrawn for analysis of input protein levels (load sample). Beads were prewashed with lysis-buffer (100 µL) before incubated with protein extract (300 µL at 1 ug/µL) at four degrees overnight. Beads were pelleted by centrifugation at 1000 rpm, 1 min, washed three times with lysis buffer and boiled at 95℃, 5 min with Laemmli buffer (IP sample). 10 µL of each sample was

separated on an SDS-PAGE gel for 1.5 hr at 120V an blotted as described below. Membranes were incubated overnight with the primary antibody at 4 degrees.

## Immunoblot analysis

Immunoblot analysis of selected proteins was performed as described previously (*Biteau et al., 2003*; *Molin et al., 2011*). Prior to separation on 12% Bis-Tris NuPAGE gels using an XCell SureLock MiniCell (Invitrogen) in NuPAGE MOPS running buffer as recommended by the supplier protein extracts were heated in Laemlii buffer (pH 8.7) either in the presence of β-mercaptoethanol (5%, reducing) or not (non-reducing) as indicated. Transfer to Immobilon-FL PVDF membranes was done using an XCell II Blot Module kit. Membranes were analyzed by the Odyssey infrared imaging system (LI-COR biosciences) as recommended by the suppliers.

Glutathionylation of Tpk1 was assayed using anti-glutathione immunoblot on Tpk1-HB immuno-precipitated by $Ni^{2+}$-Sepharose beads following a simplified protocol similar to that used during MS sample preparation (see below). We verified that the anti-glutathione immunoblot signal in Tpk1 completely disappeared upon extract reduction by β-mercaptoethanol.

## Growth conditions for MS analysis

Cells were grown at 30°C in yeast extract/peptone (YP) medium, containing 2% glucose as carbon source. Three independent experimental replicates were performed for each experimental condition. For each replicate, we inoculated 750 ml YPD cultures, which were incubated (with shaking) overnight until OD600 = 1. Oxidative stress was induced by adding 0.4 mM or 0.8 mM (final concentration) $H_2O_2$ for 10 min.

## Mass spectrometric sample preparation

HB (poly histidine, biotinylation signal) tandem affinity purifications were performed as described elsewhere (*Reiter et al., 2012*). Cells were harvested by filtration and immediately deep-frozen in liquid $N_2$. Cells were grinded using a SPEX Freezer Mill 6870 (SPEXSamplePrep, Metuchen, NJ, USA) with the following settings: 7 cycles: 3 min breakage (15 CPS), 3 min cooling, resuspended in buffer 1 (6 M guanidine HCl, 50 mM Tris pH8.0, 5 mM NaF, 1 mM PMSF, 0.1% Tween, cOmplete Protease inhibitor cocktail, pH 8) and cleared of debris by centrifugation 13.500 x g, 15 min, 4°C. Cleared extracts were incubated (4 hr, room temperature) with $Ni^{2+}$-Sepharose beads, washed with urea buffer (8M urea, 50 mM sodium phosphate buffer pH8.0, 300 mM NaCl, 0.1% Tween20) and urea buffer pH 6.3. Proteins were eluted in urea buffer pH 4.3 containing 10 mM EDTA, incubated overnight with streptavidin-agarose beads, washed using urea wash buffer containing 1% SDS and urea wash buffer without SDS. Beads were washed five times with 50 mM ammonium bicarbonate (ABC). Cys-residues were alkylated with IAA (25% w/w of the estimated amount of protein). Excess IAA was washed out by ABC. Proteins were digested with 300 ng trypsin at 37°C overnight. Digestion was stopped with trifluoroacetic acid (0.5% final concentration) and the peptides were desalted using C18 Stagetips (*Rappsilber et al., 2007*). 50 fmol of the Peptide Retention Time Calibration Mixture was spiked in each sample for quality control.

## Mass spectrometry analysis of Tpk1

Peptides were separated on an Ultimate 3000 RSLC nano-flow chromatography system (Thermo-Fisher), using a pre-column (Acclaim PepMap $C_{18}$, 2 cm ×0.1 mm, 5 µm, Thermo-Fisher), and a $C_{18}$ analytical column (Acclaim PepMap C18, 50 cm ×0.75 mm, 2 µm, Thermo-Fisher). A segmented linear gradient from 2% to 35% solvent B (solvent B: 80% acetonitrile, 0.1% formic acid; solvent A: 0.1% formic acid) was applied at a flow rate of 230 nL/min over 120 min. A Proxeon nanospray flex ion source (Thermo Fisher) using coated emitter tips (New Objective) was used for ionization. The capillary temperature was set to 200°C. Peptides were analyzed on an Orbitrap Fusion Lumos Tribrid mass spectrometer (Thermo Fisher). The mass spectrometer was operated in data-dependent mode, survey scans were obtained in a mass range of 380–1500 m/z with lock mass activated, at a resolution of 120,000 at 200 m/z and an automatic gain control (AGC) target value of 4E5. The maximum cycle time was set to 2.5 s and the most abundant precursors were selected for fragmentation by high-energy collision at 30% collision energy. Fragmented precursors were excluded from further fragmentation for 30 s (with +/- 5 ppm accuracy) and peptides with charge +one or > +six were

excluded from MS/MS analysis. The most abundant Tpk1 Cys containing peptide forms have been added to an inclusion list as specified in the raw files. MS proteomics data have been deposited to the ProteomeXchange Consortium through the Proteomics Identifications database (PRIDE) partner repository (*Vizcaíno et al., 2016*) with the data set identifiers PXD012617.

### Closed database search

Peptide identification and label free quantification (LFQ) were performed using MaxQuant (version 1.6.0.16) with default parameters. *Saccharomyces cerevisiae* reference proteome database (UniProt, version January 2017) in combination with a common laboratory contaminants database (MQ) was used for peptide spectrum matching. N-terminal acetylation, deamidation of asparagine and glutamine, oxidation of methionine, tri-oxidation and glutathionylation of cysteine and phosphorylation of serine, threonine and tyrosine were set as variable protein modification. Carbamidomethylation of cysteine was set as fixed. A maximum of 5 variable modifications per peptide was allowed. Leucine and isoleucine were treated as indistinguishable. Enzyme specificity was set to 'Trypsin/P'. A maximum of 2 missed cleavages per peptide was allowed. 'Requantify' and 'Match between runs' was activated. MaxLFQ (implemented in the MaxQuant package) was used for MS1-based label free quantification and normalization of protein groups.

### Open database search of selected peptides

To screen for protein modifications in an unbiased manner we initially performed an open search using MSFragger in FragPipe (*Kong et al., 2017*). The default open search parameters were used, with trypsin specificity, +/- 500 Da windows and oxidation of methionine and carbamidomethylation of cysteine as variable modifications. The observed mass shifts were inspected and filtered for the most abundant and relevant modifications occurring in Tpk1.

### Targeted mass-spectrometry

Parallel-Reaction-Monitoring (PRM) assays were generated based on the peptide information obtained by MaxQuant. We selected Tpk1 peptides for targeted relative LFQ as specified in Supplementary file 1D .Peptides were separated using a 120 min gradient (HPLC setup as described above). PRM data acquisition was performed using a scheduled method with 20 min windows for each target based on the retention time determined in the shotgun-approach. Raw data were obtained on an Orbitrap Q Exactive HF-X (Thermo Fisher Scientific) mass spectrometer applying the following settings: survey scan with 60 k resolution, AGC 1E6, 60 ms IT, over a range of 400 to 1400 m/z, PRM scan with 30 k resolution, AGC 1E5, 200 ms IT, isolation window of 1.0 m/z with 0.3 m/z offset, and NCE of 27%.

Wash runs were checked for potential peptide carry-over in between samples using same HPLC and MS methods. Data analysis, manual validation of all transitions (based on retention time, relative ion intensities, and mass accuracy), and relative quantification was performed in Skyline. Up to six characteristic transitions were selected for each peptide and their peak areas were summed for peptide quantification (total peak area). MS1 signals of PRTC standards were used as global standards for normalization in Skyline to account for fluctuations in instrument performance. The mean of the log2 Tpk1 non-modified peptide intensities was used to normalize Tpk1 modified peptides and Tsa1 peptides to account for differences in Tpk1 levels. Tsa1 peptide intensities (anti-log) were summed up to obtain values for relative protein abundance.

### Cysteine sulfenylation assay by DYn-2 labeling, protein extraction and click chemistry

Mid-exponential cells (10 ml at $OD_{600}$ = 0.5) were treated with of DYn-2 (0.5 mM) for 30 min, at 30°C and cell suspensions were next exposed to 0.5 mM $H_2O_2$ for 5 min. To the cultures trichloroacetic acid (TCA) was added to a final concentration of 20%, followed by centrifugation (6000 x g, 5 min, 4°C) and pellets were lysed with glass beads (equivalent of 0.1 ml of beads) in 0.2 ml of TCA (20%). Lysates were centirfuged (14000 x g, 15 min, 4°C) and pellets were washed twice with acetone, dried and solubilized in 0.2 ml Hepes (100 mM) buffer containing cOmplete mini EDTA-free protease inhibitor cocktail (Roche) (one tablet/20 ml of buffer solution), 25 μg/ml phenylmethylsulfonylfluoride, 0.1% Nonidet P-40, 2% SDS, pH 7.4. Protein content was determined using a standard DC

Protein Assay (Bio-Rad). A copper (I)-catalyzed azide-alkyne cycloaddition (CuAAC) click chemistry reaction was performed on 0.2 mg of protein as previously described (*Truong and Carroll, 2012*; *Yang et al., 2015*). Briefly, cyanine5 azide (0.5 mM), copper(II)-TBTA complex (1 mM) and ascorbate (2 mM) were added to the lysates, protected from light and incubated for 1 hr at room temperature under rotation. The CuAAC reaction was quenched by adding EDTA (1 mM) for 10 min. The solution was precipitated by methanol/chloroform precipitation (sample/methanol/chloroform, 4/4/1 (v/v/v)) and centrifuged (14000 x g, 15 min, 4°C). The protein pellet obtained were between the organic and aqueous layers, both layers were aspirated. A solution of methanol/chloroform ($H_2O$/methanol/chloroform, 4/4/1 (v/v/v) was added to the protein pellet and centrifuged (14000 x g, 15 min, 4°C). Both layers were aspirated and the obtained pellet was subsequently washed twice with methanol. Protein pellets were resuspended in 100 mM Hepes buffer containing 2% SDS. Biotinylated proteins were enriched with Pierce streptavidin bead (Thermo Scientific). The protein pellets were mixed to a pre-washed streptavidin beads (100 mM Hepes buffer). The samples were incubated for 2 hr at room temperature and subsequently washed twice with 1% SDS, twice with 4M urea, once with 1M NaCl and twice with PBS. After each wash step, beads were collected by centrifugation. Beads were finally resuspended in 5X Laemmli buffer and boiled for 5 min at 95°C. Samples were resolved by SDS-PAGE and analyzed for fluorescence at 700 nm (Cyanine5) on an Odyssey CLx (Licor).

## Homology modeling

A model of the yeast PKA tetramer structure was obtained by homology modeling. The protein sequences of yeast Tpk1 (catalytic subunit of PKA) and Bcy1 (regulatory subunit of PKA) were obtained from Genbank (ID: 1023942850 and ID: 1023943330, respectively). The crystal structure of mouse PKA (PDBID: 3TNP) was used as the template for the homology calculations. The catalytic and regulatory subunits of yeast PKA and mouse PKA shares 48% and 42% sequence similarity, respectively. The homology model was built using StructurePrediction panel (*Jacobson et al., 2002*) in Schrödinger Suite (Schrödinger, LLC, New York, NY). The ClustralW method was used to align the target and template sequences in Prime, the energy-based was selected for model building method, and homo-multimer was selected for multi-template model type.

## Covalent docking

Covalent docking was carried out to obtain a model for glutathionylated Tpk1. The Tpk1 crystal structure (PDB ID: 1FOT, *Mashhoon et al., 2001*) were prepared using the Protein Preparation utility in Schrodinger to assign the correct protonation state and fix the missing side chains and loops. The glutathione was built by 3D builder and prepared by LigPre utility in Schrodinger. The Covalent-Dock panel (*Zhu et al., 2014*) in Schrodinger was used to predict the pose of the glutathione attaching to Cys243. The reaction type was set to be disulfide formation, the docking mode was set to be thorough pose prediction, the other parameters were all set to be default. At the final step, Prime Energy was used to rank the poses of the ligand. Covalent docking was performed on dephosphorylated Tpk1 structure.

## Molecular dynamics simulations

Molecular dynamics simulations were carried out to study structural changes of Tpk1 upon phosphorylation and glutathionylation. MD simulations non-modified Tpk1, Cys243 glutathionylation Tpk1, Thr241 phosphorylation Tpk1, Cys243 glutathionylation and Tpk1 phosphorylation co-existed Tpk1 were carried out. The GROMACS software (*Abraham et al., 2015*) was used for the MD simulations and the Amber 99 (*Ponder and Case, 2003*) force field was selected to assign the parameters for different amino acid residues. The glutathionylation and phosphorylation parameters was generated from Ambertools, and incorporated into the GROMACS software.

The systems were solvated with a buffer distance of 10.0 Å TIP3P water in periodic boxes, and then 0.1 mol/L of Na+ and Cl− ions were added to adjust the systems to electroneutrality condition. Then 200 steps of the steepest descent energy minimization was carried out to remove close contacts in the obtained systems. A 2ns position-restrained simulation with a constant pressure ensemble (NPT) was performed to make sure the water molecules would reach more favorable positions. The parameters for position-restrained simulation are set to be: a time step = 1 fs, temperature = 298 K, and coupling pressure = 1 bar, Coulomb cutoff = 10 Å, Lennard-Jones cutoff = 10 Å, particle-

mesh Ewald summation (*Darden et al., 1993*; *Essmann et al., 1995*) was used for longrange electrostics. The temperature and pressure was controlled by Berendsen coupling algorithm (*Berendsen et al., 1984*), with the time constants of 0.1 ps for temperature and 1.0 ps for pressure coupling. All bond lengths were contrained by the LINCS algorithm (*Hess et al., 1997*). Following the position-restrained simulation, 100 ns production simulations with NPT ensemble were performed on each system for further study the protein conformational changes. In this step, the Nose−´ Hoover thermostat (*Hoover, 1985*), with a time constant 0.1 ps, was used to control the temperature and the Parrinello−Rahman barostat (*Parrinello and Rahman, 1981*), with a time constant 1.0 ps, was used to control the pressure. The other parameters were the same as those in the position-restrained simulations.

## Quantification and statistical analysis

All experiments were repeated at least three times (biological replicates) to ensure reproducibility. Biological replicates of experiments were performed in separate, independent experiments (typically on a separate day). No data were excluded in averages/median values presented in figures. Details on the number of replicates and statistical analyses performed in relation to the specific figures are available below.

*Figure 1* (B) Lifespans were tested for statistical significance by the Mann-Whitney U test (www.socscistatistics.com/tests/mannwhitney/Default2.aspx). B) Lifespans of wt control and o/e *TSA1* strains are significantly different using the Mann Whitney U test (n = 167 and 168 cells, p<0.00001). Lifespans of *pde2Δ* control and *pde2Δ* o/e *TSA1* strains are not significantly different (n = 81 and 84, respectively, p=0.58). D) Hsp12 levels are significantly different between control and o/e *TSA1* strains (n = 3, p=0.033) whereas Act1 levels are not (n = 3, p=0.69). E) Lifespans of the wt (n = 157) vs the *tsa1Δ* (n = 293) and the *tsa1Δ* vs *ras2Δtsa1Δ* (n = 283) are significantly different at p<0.00001. The lifespan of the *ras2Δ* (n = 138) is not significantly different from the *ras2Δtsa1Δ* (p=0.276). F) Lifespans of the wt (n = 157) vs the *pde2Δ* (n = 120), the *ras2Δtsa1Δ* vs *ras2Δpde2Δtsa1Δ* (n = 164) and the *tsa1Δ* (n = 293) vs *pde2Δtsa1Δ* (n = 242) are significantly different at p<0.00001. Lifespans of the *ras2Δpde2Δ* (n = 124) vs *pde2Δ* are significantly different (p=0.00068) whereas the lifespans of *pde2Δ* vs *pde2Δtsa1Δ* are not significantly different (p=0.757).

*Figure 2* (A) Doubling times of wt and *ras2Δ* strains are significantly different at p=0.047 whereas the difference between the *tsa1Δ* and the *ras2Δtsa1Δ* is not statistically significant using a two-sided t-test assuming equal variance (p=0.77). C) Doubling times of control and mc-*IRA2* strains are significantly different for the wt (n = 7 each, p=7.4×10$^{-6}$), the *tsa1ΔYF* (n = 3 and 4, respectively, p=0.0032), *msn2Δmsn4Δ* (n = 3 each, p=0.026) and *trx1Δtrx2Δ* (n = 15 and 13, respectively, p=0.012). In none of the other strains are control and mc-*IRA2* different (*tsa1Δ* n = 3 each, p=0.87; *tsa1C48S*, n = 3 each, p=0.71; *tsa1C171S*, n = 4 each, p=0.11; *tsa1ΔYFC171S*, n = 4 each, p=0.77; *pde2Δ*, n = 3 each, p=0.66). D) Relative *HSP12* levels were significantly different between wt control and mc-*IRA2* strains (n = 15 and 9, respectively, p=1.0×10$^{-14}$), between wt mc-*IRA2* and *tsa1Δ* mc-*IRA2* strains (n = 9 and 8, respectively, p=1.9×10$^{-6}$), between wt mc-*IRA2* and *tsa1C171S* mc-*IRA2* strains (n = 9 and 4, respectively, p=0.026), between wt mc-*IRA2* and *tsa1ΔYFC171S* mc-*IRA2* strains (n = 9 and 6, respectively, p=0.00083) and between wt mc-*IRA2* and *pde2Δ* mc-*IRA2* strains (n = 9 and 6, respectively, p=4.8×10$^{-8}$). No significant difference was seen between wt mc-*IRA2* and *tsa1-ΔYF* mc-*IRA2* strains (n = 9 and 3, respectively, p=0.53). Relative *CTT1* levels were significantly different between wt control and mc-*IRA2* strains (n = 24 and 21, respectively, p=3.4×10$^{-13}$), between wt mc-*IRA2* and *tsa1Δ* mc-*IRA2* strains (n = 21 and 6, respectively, p=0.0073), between wt mc-*IRA2* and *tsa1C171S* mc-*IRA2* strains (n = 9 and 4, respectively, p=0.026), between wt mc-*IRA2* and *tsa1ΔYF* mc-*IRA2* strains (n = 21 and 3, respectively, p=0.027), between wt mc-*IRA2* and *tsa1ΔYFC171S* mc-*IRA2* strains (n = 21 and 6, respectively, p=p = 4.9 x 10$^{-5}$) and between wt mc-*IRA2* and *pde2Δ* mc-*IRA2* strains (n = 21 and 6, respectively, p=3.5×10$^{-7}$). F) Doubling times of control and mc-*IRA2* strains are significantly different for the wt control (n = 3 each, p=0.00042), the wt o/e *PDE2* (n = 3 each, p=0.00091) and for the *tsa1Δ* mc-*PDE2* strain (n = 3, p=0.0058) but not for the *tsa1Δ* control strain (n = 3 each, p=0.20). G) The time Msn2 spent in the nucleus is significantly different in the wt vector control (n = 82) vs. *mc-BCY1* (n = 76, p<0.001) but not *tsa1Δ* vector control (n = 46) vs. *mc-BCY1* (n = 74, p=0.14). H) Relative Ras2-GTP/total Ras values in the control and mc-*IRA2* are significantly different in a two tailed t-test with unequal variance for the wt (n = 3, p=0.0041). Values for the *pde2Δ* control vs mc-*IRA2* (n = 3, p=0.015) and the *tsa1Δ* control vs mc-*IRA2* (n = 3, p=0.030)

are significantly different in a two-tailed t-test with equal variance. I) cAMP levels are significantly different only between wt and $pde2\Delta$ strains (n = 4 each, wt yEP24 vs pde2 yEP24, p=0.0050 and wt pKF56 vs pde2 pKF56, p=$1.4\times10^{-5}$). No significant differences were seen between wt and $tsa1\Delta$ strains (n = 4 each, wt yEP24 vs $tsa1\Delta$ yEP24, p=0.86 and wt pKF56 vs $tsa1\Delta$ pKF56 p=0.47) or between the wt yEP24 and wt pKF56 (p=0.13).

*Figure 3* (A) Lifespans of the wt (n = 168) vs the $tsa1\Delta$ mutant (n = 293), wt vs $tsa1C48S$ (n = 70), wt vs $tsa1C171S$ (n = 120), $tsa1\Delta YF$ (n = 255) vs $tsa1\Delta YFC171S$ (n = 70) are all different at p<0.00001. The lifespan of the $tsa1\Delta YF$ mutant is different from the wt at p<0.00854 whereas no significant difference was seen between the $tsa1\Delta$ vs $tsa1C48S$ (p=0.11), $tsa1C48S$ vs $tsa1C171S$ (p=0.31), $tsa1\Delta YFC171S$ vs $tsa1C171S$ (p=0.23). C. $H_2O_2$ resistance is significantly different between wt and $ras2\Delta$ strains (p=0.013), wt and $tsa1\Delta$ (p=0.0049) and $tsa1\Delta$ and $ras2\Delta tsa1\Delta$ (p=0.010). D. $H_2O_2$ resistance is significantly different between wt control and o/e $TSA1$ strains (p=0.0085), between wt o/e $TSA1$ and $pde2\Delta$ o/e $TSA1$ strains (p=0.0082) but not between $pde2\Delta$ control and o/e $TSA1$ strains (p=0.56). E. $H_2O_2$ resistance is significantly different between wt vector and mc-$IRA2$ strains (p=0.016), wt vector and $tsa1\Delta$ vector strains (p=0.049), $tsa1\Delta$ vector and mc-$IRA2$ strains (p=0.00056), $tsa1\Delta$ mc-$IRA2$ and $pde2\Delta tsa1\Delta$ mc-$IRA2$ strains (p=0.0025) but neither the $pde2\Delta$ vector and mc-$IRA2$ strains (p=0.40) nor $trx1\Delta trx2\Delta$ vector and $trx1\Delta trx2\Delta$ mc-$IRA2$ strains (p=0.24). F. The scavenging rates of the wt and the $tsa1\Delta$ mutant following the addition of 0.4 mM are not significant in a two-tailed t-test assuming equal variance (p=0.684). G. Fluorescence ratios 500/420 nm of the HyPer3 expressing strains are significantly different between the wt young (n = 231) vs old (n = 319) (p=$2.42\times10^{-13}$) and wt young vs wt young +$H_2O_2$ (n = 202) (p=$5.27\times10^{-76}$) but not when comparing wt old vs $tsa1\Delta$ old (n = 236)(p=0.101). H. Fluorescence ratios 500/420 nm of the HyPer3 expressing strains are neither significantly different between the wt young (n = 404) vs the o/e $TSA1$ young (n = 579, p=0.069) nor the wt old (n = 190) vs o/e $TSA1$ old (n = 204, p=0.755).

*Figure 4* (C) The abundances of all the three T241-phosphorylated peptides decreased significantly upon adding either 0.4 mM or 0.8 mM H2O2 (for the C243-SH peptide p=0.05 and 0.037 respectively, for the C243-SSG peptide p=0.015 and 0.025 respectively whereas for the C243-$SO_3$H peptide p=0.011 and 0.0049, respectively. The quantity of the C243-SH T241 non-modified peptide did not change significantly upon the addition of 0.4 and 0.8 mM $H_2O_2$ (p=0.20 and 0.54, respectively) whereas the C243-SSG T241 non-modified peptide increased significantly following 0.4 mM (p=0.038) but not at 0.8 mM (p=0.17). F. Tpk1-S-SG levels are significantly different between wt with and without $H_2O_2$ (p=0.012), but not between wt and $tsa1\Delta$ without $H_2O_2$ (p=0.453) or in the $tsa1\Delta$ with and without $H_2O_2$ (p=0.264).

*Figure 5* (A) $H_2O_2$ resistance is significantly different between wt pRS313 vector control and $tpk2\Delta tpk3\Delta$ pRS313 vector control strains (p=0.030), $tpk1\Delta tpk2\Delta tpk3\Delta$ $pTPK1$ and $ptpk1C243A$-strains (p=0.030), $tpk1\Delta tpk2\Delta tpk3\Delta$ $pTPK1$ and $ptpk1T241A$ strains (p=0.0020) but not $tpk2\Delta tpk3\Delta$ pRS313 and $tpk1\Delta tpk2\Delta tpk3\Delta$ $pTPK1$ strains (p=1.00). (B) $H_2O_2$ resistance is significantly different between control $pTPK1$ and $ptpk1C243A$ strains (p=0.043), control $pTPK1$ and $pTSA1$ $pTPK1$ strains (p=0.0072), $pTSA1$ $pTPK1$ and $ptpk1C243A$ strains (p=0.0014) but not between control $ptpk1C243A$ and $pTSA1$ $ptpk1C243A$ strains (p=0.064). (C) $H_2O_2$ resistance is significantly different between $TSA1$ $pTPK1$ and $ptpk1T241A$ strains (p=0.022), $TSA1$ $pTPK1$ and $tsa1\Delta$ $pTPK1$ strains (p=0.031), $tsa1\Delta$ $pTPK1$ and $ptpk1T241A$ strains (p=0.013) but not between $TSA1$ and $tsa1\Delta$ $ptpk1T241A$ strains (p=0.090).

## Acknowledgements

We are grateful to Karin Voordeckers, Joseph Heitman and Robert J Deschenes for reagents and Mattias Johansson and Lainy Ramirez for technical assistance.

## Additional information

### Funding

| Funder | Grant reference number | Author |
| --- | --- | --- |
| Cancerfonden | 2017-778 | Mikael Molin |

| Vetenskapsrådet | NT 2011-5170 | Mikael Molin |
| Stiftelsen Olle Engkvist Byggmästare | 20120620 | Mikael Molin |
| Carl Tryggers Stiftelse för Vetenskaplig Forskning | CTS 11:306 | Mikael Molin |
| Carl Tryggers Stiftelse för Vetenskaplig Forskning | CTS 13:277 | Mikael Molin |
| Agence Nationale de la Recherche | PrxAge | Michel B Toledano |
| Agence Nationale de la Recherche | ERRed2 | Michel B Toledano |
| Vetenskapsrådet | NT 2019-03937 | Thomas Nyström |
| Knut och Alice Wallenbergs Stiftelse | 2017-0091 | Thomas Nyström |
| Knut och Alice Wallenbergs Stiftelse | 2015-0272 | Thomas Nyström |

The funders had no role in study design, data collection and interpretation, or the decision to submit the work for publication.

## Author contributions
Friederike Roger, Investigation, Writing - original draft, Writing - review and editing; Cecilia Picazo, Marouane Libiad, Chikako Asami, Sarah Hanzén, Gilles Lagniel, Investigation; Wolfgang Reiter, Methodology, Writing - original draft; Chunxia Gao, Investigation, Methodology; Niek Welkenhuysen, Data curation, Methodology; Jean Labarre, Markus Hartl, Methodology; Thomas Nyström, Michel B Toledano, Writing - review and editing; Morten Grøtli, Supervision, Writing - review and editing; Mikael Molin, Conceptualization, Supervision, Funding acquisition, Investigation, Methodology, Writing - original draft, Project administration, Writing - review and editing

## Author ORCIDs
Wolfgang Reiter https://orcid.org/0000-0003-1266-8975
Thomas Nyström http://orcid.org/0000-0001-5489-2903
Morten Grøtli https://orcid.org/0000-0003-3621-4222
Markus Hartl http://orcid.org/0000-0002-4970-7336
Michel B Toledano https://orcid.org/0000-0002-3079-1179
Mikael Molin https://orcid.org/0000-0002-3903-8503

## Decision letter and Author response
Decision letter https://doi.org/10.7554/eLife.60346.sa1
Author response https://doi.org/10.7554/eLife.60346.sa2

# Additional files
## Supplementary files
• Supplementary file 1. Mass spectrometry data.

• Transparent reporting form

## Data availability
Proteomics data have been deposited in the PRIDE repository.

The following dataset was generated:

| Author(s) | Year | Dataset title | Dataset URL | Database and Identifier |
| --- | --- | --- | --- | --- |
| Roger F, Picazo C, | 2019 | Peroxiredoxin promotes longevity | https://www.ebi.ac.uk/ | PRIDE, PXD012617 |

| Reiter W, Libiad M, Asami C, Hanzén S, Gao C, Lagniel G, Welkenhuysen N, Labarre J, Nyström T, Grøtli M, Hartl M, Toledano M, Molin M | and H2O2-resistance in yeast through redox modulation of protein kinase A | pride/archive/projects/ PXD012617 |
|---|---|---|

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
