## [Decision Letter]

**Acceptance summary:**

The manuscript is a beautiful example of yeast genetics – shedding mechanistic light on a highly conserved and extremely important response pathway affecting life- and health-span. Specifically it uncovers the mechanism by which the peroxiredoxin Tsa1 improves longevity and H_2_O_2_ tolerance. The exciting finding is that Tsa1 promotes

both H_2_O_2_ tolerance and lifespan extension not through its well-studied role as an H_2_O_2_ cavenger, but by signaling towards PKA. These findings create a much richer view of peroxiredoxins – studied for over 25 years – showing that these enzymes have important roles beyond their H_2_O_2_ scavenging function.

**Decision letter after peer review:**

[Editors’ note: the authors submitted for reconsideration following the decision after peer review. What follows is the decision letter after the first round of review.]

Thank you for submitting your work entitled "Peroxiredoxin promotes longevity and H_2_O_2_-resistance in yeast through redox-modulation of protein kinase A" for consideration by *eLife*. Your article has been reviewed by three peer reviewers, one of whom is a member of our Board of Reviewing Editors, and the evaluation has been overseen by a Senior Editor. The reviewers have opted to remain anonymous.

Our decision has been reached after consultation between the reviewers. Based on these discussions and the individual reviews below, we regret to inform you that your work will not be considered further for publication in *eLife*. Please find the combined reasons for this as well as suggestions for improvements for future submissions below.

Combined Reviews:

The manuscript by Roger et al. aims to identify specific targets of nutrient signaling explaining the beneficial effects of Caloric Restriction (CR). They focus on the yeast peroxiredoxin Tsa1, which they have previously shown is stimulated by CR and required for the increase in both H_2_O_2_ tolerance and lifespan. They set out to uncover whether Tsa1 performs this role by reducing H_2_O_2_ levels or rather by acting as a signaling molecule. They suggest that in the context of CR, Tsa1 works through its function on PKA, which it regulates by modifying a conserved cysteine in the most highly expressed PKA catalytic subunit.

In our eyes the authors provide compelling evidence that Tsa1's overexpression phenotype of lifespan extension and recovery of cells from H_2_O_2_ challenges is because of its ability to regulate Protein Kinase A activity. The finding that Tsa1 mediates its effect through PKA identifies a previously unappreciated mechanism of redox state regulation of PKA. Interestingly, while the authors work is in yeast, the sites of modification on PKA that are mediated by Tsa1 are conserved. This suggests that this mechanism of action is also conserved and should impact how the aging and redox fields consider ROS and its effect on aging.

However, there were some major concerns raised that lead us to reject this manuscript:

1) The function of Tsa1 through PKA was already partially shown in Bodvard et al., 2017.

2) The manuscript claims that Tsa1 modulates the redox state of Tpk1 cysteines, however, no direct evidence is provided. There is no data that would demonstrate a Tsa1-Tpk1 interaction. It is not clear what kind of redox mechanism is actually proposed by the authors. Tsa1 directly oxidizing Tpk1? What is the role of Trxs? How does glutathione fit in here? There is also no discussion addressing these issues.

3) There doesn't seem to be a clear experiment that properly shows that the absence of Tsa1 results in a different Tpk1 redox state. The absence of Tsa1 does not seem to affect Tpk1 glutathionylation in the absence of exogenously applied H_2_O_2_ (Figure 6I). The PEGylation experiments were done with the addition of exogenous H_2_O_2_, and PEGylation was still observed with the C243A mutant. The connection between glutathionylation of a protein and peroxiredoxin activity (which is normally coupled to the Trx system and not to the GSH/Grx system), if true, remains very much unclear.

4) The only piece of evidence suggesting that the cysteine residues of Tpk1 are important seems to be Figure 6G. Here, and only within the context of the *tpk1Δtpk2Δtpk3Δ* triple deletion, the authors show a lower resistance to H_2_O_2_. Like in some other places, the information about the specifics of the experiments seem to be incomplete. Here, it is not clear how resistance was determined. Is it from a spot test? Is it from a viability assay? How much H_2_O_2_ was added? What were the conditions? There is no control showing that Tpk1 C243A is expressed at the same level as Tpk1wt.

5) Although a lot of weight is put on the determination that Tsa1 acts independently of H_2_O_2_ scavenging, it is not clear if and how the data shown in Figure 1 can actually demonstrate this. Cells lacking individual thiol peroxidases usually adapt to adjust their overall reductive capacity, thus it is not surprising that they do not exhibit differences in the overall H_2_O_2_ uptake rate or in the average redox state of cytosolic Hyper3. But redox control can be highly localized and Tsa1 may still locally control exposure of Tpx1 to H_2_O_2_ by acting as a scavenger. The involvement of Trxs in the process would rather speak in favor of a scavenging scenario.

6) Finally, some of the data shown in Figure 4A seem to be taken from a data set that was already published: Figure 1A in Hanzen et al., 2016; this is not explicitly indicated as far as I can see.

---

## [Author Response]

Combined Reviews:The manuscript by Roger et al. aims to identify specific targets of nutrient signaling explaining the beneficial effects of Caloric Restriction (CR). They focus on the yeast peroxiredoxin Tsa1, which they have previously shown is stimulated by CR and required for the increase in both H_2_O_2_ tolerance and lifespan. They set out to uncover whether Tsa1 performs this role by reducing H_2_O_2_ levels or rather by acting as a signaling molecule. They suggest that in the context of CR, Tsa1 works through its function on PKA, which it regulates by modifying a conserved cysteine in the most highly expressed PKA catalytic subunit.In our eyes the authors provide compelling evidence that Tsa1's overexpression phenotype of lifespan extension and recovery of cells from H_2_O_2_ challenges is because of its ability to regulate Protein Kinase A activity. The finding that Tsa1 mediates its effect through PKA identifies a previously unappreciated mechanism of redox state regulation of PKA. Interestingly, while the authors work is in yeast, the sites of modification on PKA that are mediated by Tsa1 are conserved. This suggests that this mechanism of action is also conserved and should impact how the aging and redox fields consider ROS and its effect on aging.However, there were some major concerns raised that lead us to reject this manuscript:1) The function of Tsa1 through PKA was already partially shown in Bodvard et al., 2017.

We indeed previously showed that Tsa1 impacts on PKA-dependent phosphorylation of the transcription factor Msn2 by H_2_O_2_, which indirectly made inferences on the activity of PKA. However, the focus of the two studies is totally different: the Bodvard et al. study enquired about the activation of the Msn2 transcription factor in response to light and H_2_O_2_, whereas in the current paper, emphasis is on longevity, H_2_O_2_ tolerance, only using the Msn2 response as readout of PKA activity amongst many others. Furthermore, the main conclusions reached in the new study are totally new and provocative, as described below, also addressing for the first time the mechanism by which Tsa1 represses PKA:

1) Our main conclusion that Tsa1 promotes yeast H2O2 tolerance and lifespan extension not as an H_2_O_2_ scavenger, but by signaling towards PKA (see below) is new and a highly provocative concept, shaking up ~25 years of research on peroxiredoxins since their discovery indicating that these enzymes promote H_2_O_2_ tolerance by their enzymatic H_2_O_2_ scavenging function.

2) We provide genetic and mechanistic information on this regulation both regarding the exact target of Tsa1 in the PKA pathway, PKA itself, and the Tsa1 biochemical function effecting this response. In the previous study we only indirectly inferred of the impact of Tsa1 on PKA through the specific pattern of Msn2 phosphorylation.

3) Furthermore, we show how the identified Tsa1 biochemical function interferes with PKA, by signaling through its interaction with a PKA catalytic subunit (Tpk1) and promotion of PKA Cys residue sulfenylation in response to H_2_O_2_.

4) We also identify a Tpk1 Cys residue in the kinase activation loop that is conserved in all eukaryotic PKA enzymes, and is required for Tsa1-mediated H_2_O_2_-resistance.

2) The manuscript claims that Tsa1 modulates the redox state of Tpk1 cysteines, however, no direct evidence is provided. There is no data that would demonstrate a Tsa1-Tpk1 interaction. It is not clear what kind of redox mechanism is actually proposed by the authors. Tsa1 directly oxidizing Tpk1? What is the role of Trxs? How does glutathione fit in here? There is also no discussion addressing these issues.3) There doesn't seem to be a clear experiment that properly shows that the absence of Tsa1 results in a different Tpk1 redox state. The absence of Tsa1 does not seem to affect Tpk1 glutathionylation in the absence of exogenously applied H_2_O_2_ (Figure 6I).

The previous submission provided data showing that Tsa1 regulates Tpk1 deglutathionylation. In the current, revised submission, we now provide evidence of a (i) direct interaction of Tsa1 and Tpk1 by coimmunoprecipitation and (ii) that Tpk1 becomes sulfenylated in response to H_2_O_2_ in a manner dependent on Tsa1. (iii) We also confirm that Tpk1 is glutathionylated, and that this modification decreases with H_2_O_2_, although we have so far failed modeling the redox mechanism modulating PKA. (iv) We have also thoroughly addressed the role of thioredoxins, both genetically and biochemically, and came to the conclusion that these enzymes are only very partially involved in PKA repression, which departs from the Bodvard et al. study, again indicating that although PKA is a target of repression in both studies, other signaling pathways at play differentiate these studies. (v) We are still short of a solid model of Tpk1 redox regulation, and thus have now thoroughly discussed the possible redox mechanism at play in Tpk1 repression and in particular how to reconcile Tpk1 deglutathionylation and sulfenylation in response to H_2_O_2_. Our belief is that Tpk1 sulfenylation is the driving event in Tpk1 inactivation, a question left at the appreciation of the reviewers.

The new data encompasses four new main figure panels and two figure supplement panels: Figure 4A and Figure 4—figure supplement 1A, addressing point (i) above; Figure 4D and Figure 4—figure supplement 1I, addressing point (ii) and Figures 2J and 3E addressing point (iv). We discuss these new data in subsections “Tsa1 catalytic cysteines control H2O2 resistance by repressing PKA” and “Tpk1 is sulfenylated upon H2O2 addition and glutathionylated on the conserved Cys243”. Relating to point (v) we updated the Discussion with a thorough review of possible redox mechanisms at play in paragraph three.

The PEGylation experiments were done with the addition of exogenous H_2_O_2_, and PEGylation was still observed with the C243A mutant.

We accept this criticism. Mal-PEG experiments are often hard to interpret and therefore we removed this experiment.

The connection between glutathionylation of a protein and peroxiredoxin activity (which is normally coupled to the Trx system and not to the GSH/Grx system), if true, remains very much unclear.

We totally agree with this statement. Our data do not suggest this eventuality, since Tpk1 glutathionylation is seen in cells lacking Tsa1, even more so than in wild type cells. We have clarified our view in the Results section, and have discussed how glutathionylation might happen in the context of our new data showing Tpk1 sulfenylation.

4) The only piece of evidence suggesting that the cysteine residues of Tpk1 are important seems to be Figure 6G. Here, and only within the context of the tpk1Δtpk2Δtpk3Δ triple deletion, the authors show a lower resistance to H_2_O_2_. Like in some other places, the information about the specifics of the experiments seem to be incomplete. Here, it is not clear how resistance was determined. Is it from a spot test? Is it from a viability assay? How much H_2_O_2_ was added? What were the conditions? There is no control showing that Tpk1 C243A is expressed at the same level as Tpk1wt.

We take this fair criticism that we have now addressed in several ways in the revised manuscript:

1) A special section in Materials and methods describes the H_2_O_2_ spot tests experiments.

2) We have edited the figures and legends of all spot-test experiments, specifying the concentration H_2_O_2_.

3) We now provide western blot data on Tpk1 protein levels (Figure 5—figure supplement 1D).

5) Although a lot of weight is put on the determination that Tsa1 acts independently of H_2_O_2_ scavenging, it is not clear if and how the data shown in Figure 1 can actually demonstrate this. Cells lacking individual thiol peroxidases usually adapt to adjust their overall reductive capacity, thus it is not surprising that they do not exhibit differences in the overall H_2_O_2_ uptake rate or in the average redox state of cytosolic Hyper3. But redox control can be highly localized and Tsa1 may still locally control exposure of Tpx1 to H_2_O_2_ by acting as a scavenger. The involvement of Trxs in the process would rather speak in favor of a scavenging scenario.

The reviewer comment is totally correct by saying that cells adapt to the loss of a gene by many ways. However, she/he is missing the fact that the same cells that lack Tsa1 display a loss of H_2_O_2_ tolerance as seen by spot assay, a phenotype that should not be seen if indeed these cells had adapted. Therefore, these experiments can only be interpreted by confronting the results of the spot sensitivity assay against both the ability of cells to clear a bolus of H_2_O_2_ in the medium and the cellular levels of H_2_O_2_ measured with the hypersensitive HyPer3 probe.

Anyhow, that the conclusion that Tsa1 H_2_O_2_ tolerance phenotype is not a function of its scavenging activity is built on the body of evidences provided in the paper, including this experiment and the total rescue of the *tsa1Δ* H_2_O_2_ phenotype by deletion of *RAS2*.

6) Finally, some of the data shown in Figure 4A seem to be taken from a data set that was already published: Figure 1A in Hanzen et al., 2016; this is not explicitly indicated as far as I can see.

We have now replaced the data provided with the experiment-matched controls used in determining *pde2Δ* control and *pde2Δ* o/e *TSA1* life-spans.